# Podocyte-specific KLF6 primes proximal tubule CaMK1D signaling to attenuate diabetic kidney disease

Nehaben A. Gujarati[1,6], Bismark O. Frimpong ⬡ [1,6], Malaika Zaidi ⬡ [1], Robert Bronstein[1], Monica P. Revelo[2], John D. Haley[3], Igor Kravets[4], Yiqing Guo[1] & Sandeep K. Mallipattu ⬡ [1,5] ✉

Diabetic kidney disease (DKD) is the main cause of chronic kidney disease worldwide. While injury to the podocytes, visceral epithelial cells that comprise the glomerular filtration barrier, drives albuminuria, proximal tubule (PT) dysfunction is the critical mediator of DKD progression. Here, we report that the podocyte-specific induction of human *KLF6*, a zinc-finger binding transcription factor, attenuates podocyte loss, PT dysfunction, and eventual interstitial fibrosis in a male murine model of DKD. Utilizing combination of snRNA-seq, snATAC-seq, and tandem mass spectrometry, we demonstrate that podocyte-specific KLF6 triggers the release of secretory ApoJ to activate calcium/calmodulin dependent protein kinase 1D (CaMK1D) signaling in neighboring PT cells. CaMK1D is enriched in the first segment of the PT, proximal to the podocytes, and is critical to attenuating mitochondrial fission and restoring mitochondrial function under diabetic conditions. Targeting podocyte-PT signaling by enhancing ApoJ-CaMK1D might be a key therapeutic strategy in attenuating the progression of DKD.

Diabetic kidney disease (DKD) is the leading cause of chronic kidney disease (CKD) and end-stage kidney disease (ESKD) worldwide[1,2]. Endothelial injury, glomerular hypertrophy, podocyte foot process effacement, eventual podocyte loss, and the accompanying albuminuria are key features of early DKD[3], with tubular injury and interstitial fibrosis occurring in later stages of DKD[4]. While a significant proportion of individuals with diabetes develop CKD, a majority of these individuals do not progress to ESKD[4,5]. Recent studies demonstrate that while glomerular injury is the initial driver of early injury to the kidney, factors mediating proximal tubule (PT) dysfunction is the key determinant of DKD progression[6,7].

Krüppel-like factors (KLFs) are a family of zinc-finger transcription factors that play a critical role in fundamental cellular processes in multiple tissues, including the kidney, to maintain homeostasis as well as in development and in disease[8]. Among the 17 members of the KLF family, Krüppel-like factor 6 (KLF6) is an early-inducible responsive gene expressed with cell-specific diverse roles. Specifically in the podocyte, KLF6 is a key regulator of mitochondrial function and the conditional loss of *Klf6* in podocytes reduces mitochondrial complex IV assembly, which exacerbates glomerular injury in murine models of Focal Segmental Glomerulosclerosis (FSGS) and DKD[9,10]. Based on these previous studies, we sought to investigate whether podocyte-specific induction of human *KLF6* will attenuate podocyte injury in a murine model of DKD. Here, we demonstrate that this induction of *KLF6* specifically in podocytes attenuates podocyte injury and glomerulosclerosis, and improves PT injury and interstitial fibrosis under diabetic conditions in mice. While we initially suspected this improvement in PT injury was a result of reduction in podocyte injury,

[1]Division of Nephrology and Hypertension, Department of Medicine, Stony Brook University, Stony Brook, NY, USA. [2]Department of Pathology, University of Utah, Salt Lake City, UT, USA. [3]Department of Pharmacology, Stony Brook University, Stony Brook, NY, USA. [4]Division of Endocrinology, Department of Medicine, Stony Brook University, Stony Brook, NY, USA. [5]Renal Section, Northport VA Medical Center, Northport, NY, USA. [6]These authors contributed equally: Nehaben A. Gujarati, Bismark O. Frimpong. ✉e-mail: sandeep.mallipattu@stonybrookmedicine.edu

unbiased single nuclei RNA-sequencing (snRNA-seq) and Assay for Transposase-Accessible Chromatin with sequencing (ATAC-seq) demonstrate that the induction of podocyte *KLF6* under basal conditions, primes transcriptional changes in the first segment of the PT, proximal to the podocytes, through the induction of calcium/calmodulin dependent protein kinase ID (CaMK1D) signaling. We report that CaMK1D is highly enriched in the first segment of the PT and preconditions the proximal tubule against mitochondrial fission under diabetic conditions. In addition, unbiased tandem mass spectrometry showed that the induction of *KLF6* in podocytes triggers the release of secretory Apolipoprotein J (ApoJ), which subsequently undergoes cellular uptake by low-density lipoprotein-related protein 2 (Lrp2)/megalin to activate CaMK1D signaling in the PT to attenuate mitochondrial fission. To date, this is the first study to demonstrate a potential mechanism by which the podocyte preconditions the PT against injury through ApoJ-CaMK1D signaling under diabetic conditions.

## Results

### Generation and characterization of $KLF6^{PODTA}$ mice under basal conditions

Since the podocyte-specific loss of *Klf6* increases susceptibility to DKD[10], we sought to ascertain whether the podocyte-specific induction of *KLF6* conversely attenuates the progression of DKD. Mice with podocyte-specific expression of human *KLF6* (*KLF6*) were initially generated using the "tet-on" system, where the binding of chimeric tetracycline transactivator protein (rtTA) to tet-operator and gene activation only occurs in the presence of doxycycline (DOX).

Specifically, *TRE-KLF6* mice were bred with the *Podocin-rtTA* (*PODTA*) mice to generate mice with podocyte-specific expression of *KLF6* (*KLF6^PODTA*) in the setting of DOX administration. To assess the specificity of *KLF6* expression, relative *KLF6* mRNA abundance was initially measured in isolated kidney cortex, glomeruli, podocyte, and non-podocyte glomeruli fractions from *KLF6^PODTA* and *NPHS2-rtTA* mice (Supplementary Fig. 1a). While *KLF6* expression remained completely undetected in *NPHS2-rtTA* mice across glomeruli, cortex, podocyte and non-podocyte glomeruli fractions, expression in the *KLF6^PODTA* mice was primarily enriched in the podocyte and glomeruli fractions, with a lower level of expression detectable in the kidney cortex and non-podocyte glomerular fractions (Supplementary Fig. 1a). Although mouse KLF6 is expressed in both glomerular and non-glomerular cells, immunostaining for KLF6 confirmed the higher expression of podocyte-specific KLF6 in *KLF6^PODTA* mice as compared to *NPHS2-rtTA* mice (Supplementary Fig. 1b, c). In addition, *mKlf6* expression remained similar in both *KLF6^PODTA* and *NPHS2-rtTA* mice (Supplementary Fig. 1d). Furthermore, *KLF6^PODTA* mice were viable and fertile with no significant difference in albuminuria or kidney injury as compared to *NPHS2-rtTA* mice (Supplementary Fig. 1e).

### Podocyte-specific induction of *KLF6* attenuates kidney injury under diabetic conditions

To test whether the podocyte-specific induction of *KLF6* attenuates DKD, *NPHS2-rtTA* and *KLF6^PODTA* mice underwent uninephrectomy (UNx) with subsequent low-dose streptozotocin (STZ) treatment (Fig. 1a). SHAM + vehicle-treated (SHAM-VEH) *NPHS2-rtTA* and *KLF6^PODTA* mice were used as non-diabetic controls. The combination of

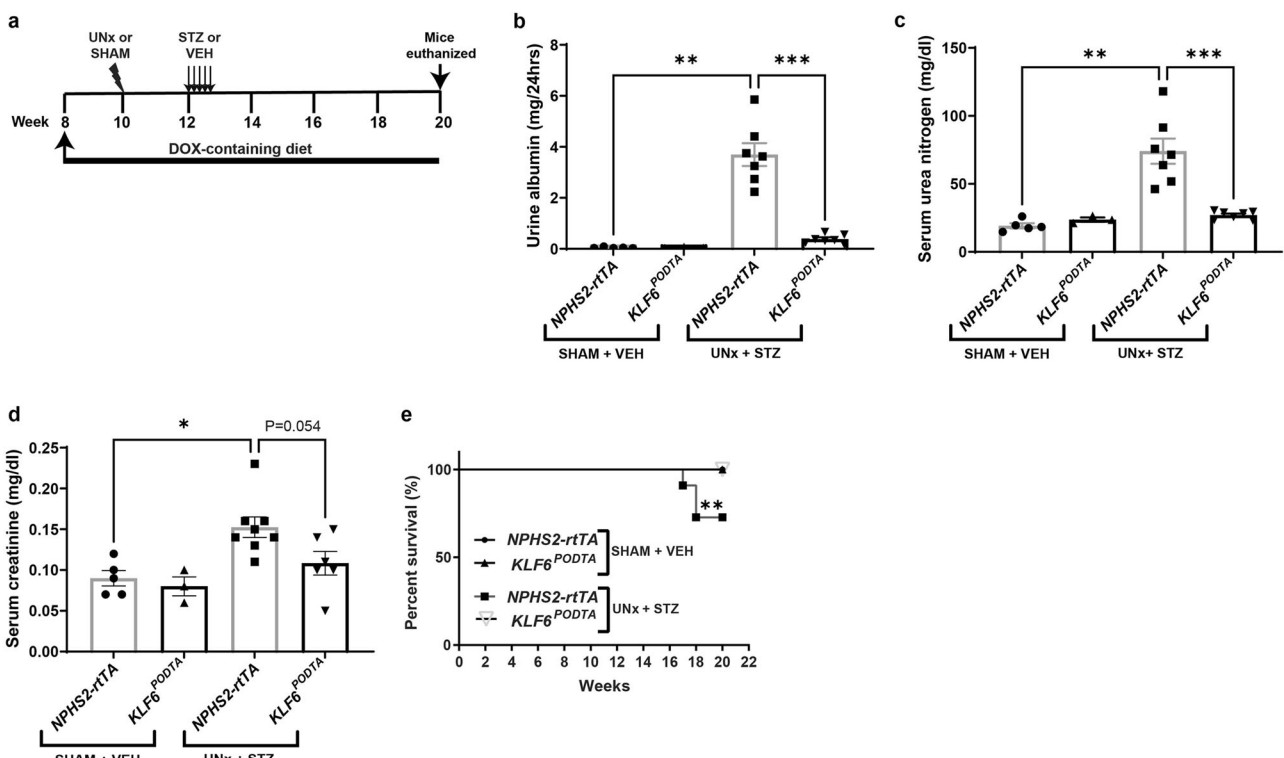

**Fig. 1 | $KLF6^{PODTA}$ mice demonstrate a reduction in albuminuria and improved kidney function and overall survival compared to *NPHS2-rtTA* mice under diabetic conditions. a** Experimental timeline for DOX treatment, UNx or SHAM procedures, STZ or VEH treatment and experimental endpoint. **b** Twenty-four hour urine albumin concentration (mg/24hrs); $p = 0.0025$ for *NPHS2-rtTA* (SHAM + VEH vs UNx + STZ) group, $p = 0.0006$ for UNx + STZ group. **c** Serum urea nitrogen (mg/dl); $p = 0.0025$ for *NPHS2-rtTA* (SHAM + VEH vs UNx+STZ), $p = 0.0006$ for UNx + STZ group. **d** Serum creatinine concentration (mg/dl) at 20 weeks of age;

$p = 0.016$ for *NPHS2-rtTA* (SHAM + VEH vs UNx + STZ), $p = 0.054$ for UNx+STZ group. For SHAM + VEH groups, $n = 5$ (*NPHS2-rtTA*), $n = 3$ (*KLF6^PODTA*) mice; for UNx + STZ groups, $n = 7$ (*NPHS2-rtTA*), $n = 6$ (*KLF6^PODTA*) mice; $^*p < 0.05$, $^{**}p < 0.01$, $^{***}p < 0.001$; Mann–Whitney test, two-sided; data presented as mean ± SEM. **e** Survival curve for all 4 groups until 20 weeks of age (for SHAM + VEH group, $n = 5$ (*NPHS2-rtTA*); $n = 3$ (*KLF6^PODTA*) mice, $n = 14$ for UNx+STZ groups; $p = 0.0094$ vs all other groups; Mantel-Cox test). Source data are provided as a Source Data file.

UNx + STZ has been shown to accelerate the development of glomerular lesions representative of DKD[11], thereby making it a suitable model to test the potential renoprotective effects of podocyte *KLF6* induction. All diabetic mice exhibited high blood glucose levels (>600 mg/dl) post-treatment, which was maintained throughout the experimental period (Supplementary Table 1). Blood glucose levels of non-diabetic groups were within the normal range throughout the experimental period (Supplementary Table 1). Kidney weights and kidney to body weight ratios were significantly increased for all diabetic groups compared to the non-diabetic groups, with no significant differences between *KLF6^PODTA^* and *NPHS2-rtTA* mice under either condition (Supplementary Table 1). While the diabetic *NPHS2-rtTA* mice exhibited an increase in albuminuria, serum urea nitrogen, and serum creatinine as compared to the non-diabetic *NPHS2-rtTA* mice and the diabetic *KLF6^PODTA^* mice, no significant differences were observed between the diabetic and non-diabetic *KLF6^PODTA^* mice (Fig. 1b–d). In addition, we observed an improvement in survival in the diabetic *KLF6^PODTA^* mice as compared to the diabetic *NPHS2-rtTA* mice (Fig. 1e). Subsequent staining with Periodic-acid Schiff (PAS) showed a significant increase in glomerular volume, mesangial expansion, % sclerotic glomeruli in all diabetic mice as compared to their respective controls (Fig. 2a–c, Supplementary Table 2). However, the diabetic *KLF6^PODTA^* mice exhibited less glomerular hypertrophy, mesangial expansion, % sclerotic glomeruli with proteinaceous casts as compared to the diabetic *NPHS2-rtTA* mice (Fig. 2a–c). Immunostaining for Wilms tumor 1 (WT1), a podocyte-specific marker, showed that the diabetic *NPHS2-rtTA* mice had fewer podocytes per glomerular area as compared to non-diabetic mice. In comparison, the podocyte number was preserved in the diabetic *KLF6^PODTA^* mice as compared to the diabetic *NPHS2-rtTA* mice (Fig. 2d). Furthermore, synaptopodin, critical for podocyte actin cytoskeleton, expression was reduced in the diabetic *NPHS2-rtTA* mice as compared to the diabetic *KLF6^PODTA^* mice and non-diabetic mice (Fig. 2e). We also investigated the ultrastructural changes using transmission electron microscopy (TEM) and found that the diabetic *NPHS2-rtTA* mice had an increase in foot process effacement and glomerular basement membrane (GBM) thickness as compared to the non-diabetic *NPHS2-rtTA* mice, which was attenuated in the diabetic *KLF6^PODTA^* mice (Fig. 2a, f, g).

Hematoxylin & eosin (H&E) with histopathological scoring by M.P.R., renal pathologist, in a blinded fashion showed an increase in interstitial inflammation in both diabetic groups as compared to their respective non-diabetic controls, with an increasing trend (non-statistically significant) in the diabetic *NPHS2-rtTA* mice as compared to the diabetic *KLF6^PODTA^* mice (Supplementary Fig. 2a, Supplementary Table 2). While both diabetic groups also had some loss of lotus lectin staining, suggesting PT brush border loss, the diabetic *NPHS2-rtTA* mice had a significant reduction as compared to the diabetic *KLF6^PODTA^* mice (Supplementary Fig. 2a, b). In addition, picrosirius red, and α-SMA staining showed an increase in staining in both diabetic groups, which was improved in the diabetic *KLF6^PODTA^* mice (Supplementary Fig. 2a, c). Interestingly, histopathological scoring noted significant tubular injury and interstitial fibrosis only in the diabetic *NPHS2-rtTA* mice (Supplementary Fig. 2a, Supplementary Table 2). Collectively, these data suggest that the podocyte-specific induction of *KLF6* attenuated glomerular and tubulointerstitial injury under diabetic conditions.

### Podocyte-specific induction of *KLF6* restores podocyte differentiation markers under diabetic conditions

To assess the potential mechanism that mediates the renoprotective effects of *KLF6^PODTA^* under diabetic conditions at the single cell level, we initially performed snRNA-seq on the kidney cortex. Rationale for choosing snRNA-seq versus scRNA-seq was based on recent studies[12]. After clearing all quality control checks, we initially generated single nuclear transcriptomes for 78,979 nuclei (Supplementary Fig. 3a, b).

Unsupervised clustering analysis subsequently identified 23 cell clusters (Supplementary Fig. 3c) and the top significant marker genes were compared to previously reported cell type markers[13] to assign cell type specific identity to these clusters. Clusters with similar cell type identity were combined to generate 18 unique clusters (Fig. 3a, b). The nuclei count for each cluster, with respect to each group was reported (Supplementary Data 1). To identify differentially expressed genes (DEGs) in the setting of podocyte-specific *KLF6* induction with and without diabetes, we initially performed differential expression analysis on all the clusters (Supplementary Data 2). Interestingly, in the podocyte cluster, there were very few DEGs (0 upregulated and 2 downregulated) in the *KLF6^PODTA^* group vs *NPHS2-rtTA* under non-diabetic conditions (Supplementary Data 2). However, under diabetic conditions, we observed a significant number of DEGs (59 upregulated and 42 downregulated) in the *KLF6^PODTA^* as compared to the *NPHS2-rtTA* mice (Fig. 3c, d). Subsequent pathway enrichment analysis on these DEGs was conducted on the podocyte cluster from the diabetic *KLF6^PODTA^* mice using *Enrichr*[14,15]. Enrichment analysis using Reactome[16], WikiPathways[17], and KEGG pathways[18] for the upregulated DEGs demonstrated key pathways involving N-linked glycosylation, axon guidance, nephrin and semaphorin interactions, as well as vascular endothelial growth factor (VEGF) signaling pathways (Fig. 3c, e). In addition, the reduced expression of podocyte structural and differentiation markers such as *Nephrin (Nphs1), Synaptopodin (Synpo), Podocalyxin (Podxl)* and *Dachshund family transcription factor 1 (Dach1)* in the diabetic *NPHS2-rtTA* mice were all significantly restored in the diabetic *KLF6^PODTA^* mice (Fig. 3c, e). Conversely, the downregulated DEGs were enriched for pathways involving focal adhesion, integrin-mediated cell adhesion, complement activation and inflammation signaling pathways (Fig. 3d, f). To determine the cell specificity of the DEGs in the podocyte cluster, we plotted the expression of these genes on a dot plot showing their expression across all the cell types (Supplementary Fig. 4a, b). We report that while the downregulated DEGs were not necessarily specific to one cell type, the upregulated genes were predominantly specific to the podocyte cluster (Supplementary Fig. 4a, b). This suggests that KLF6 might play a key role in maintenance of the mature podocyte differentiation markers in the setting of cell stress under diabetic conditions.

To explore the potential direct and indirect mechanism by which KLF6 might regulate gene expression, we initially conducted *in-silico* analysis by matching the DEGs in the podocyte cluster with previously reported KLF6 ChIP-seq expression arrays[19] obtained from the *Encyclopedia of DNA Elements (ENCODE)* project (Supplementary Fig. 5a). Class 0 DEGs were defined as having at least 1 KLF6 binding site within ±1 kb of the transcription start site (TSS), class 1 DEGs have at least 1 KLF6 binding site between ±1 and 10 kb of the TSS and class 2 DEGs as having no KLF6 binding site within ±10 kb of the TSS. Of the 59 upregulated DEGs, there were 11 class 0, 5 class 1, and 43 class 2 DEGs (Supplementary Fig. 5b), while the downregulated DEGs were 9 class 0, 2 class 1, and 31 class 2 DEGs (Supplementary Fig. 5c). To examine the contribution of the class 0 DEGs to enriched pathways, we compared the statistical significance of pathway enrichment between class 0 and class 2 DEGs (Supplementary Fig. 5d, e). Despite having fewer numbers of DEGs, upregulated class 0 DEGs were more highly enriched for N-linked glycosylation pathways, axon guidance, and nephrin interaction pathways, suggesting a potentially more direct transcriptional regulation by KLF6 (Supplementary Fig. 5d). In comparison, class 0 downregulated DEGs were enriched for pathways involving integrin and non-integrin dependent ECM interactions as compared to class 2 downregulated DEGs (Supplementary Fig. 5e).

To further characterize the potential mechanism by which podocyte-specific *KLF6* might mediate its salutary effects, we performed snATAC-seq on kidney cortex samples from *NPHS2-rtTA* and *KLF6^PODTA^* mice to ascertain the degree of chromatin accessibility in the podocyte cluster. We successfully generated chromatin accessibility

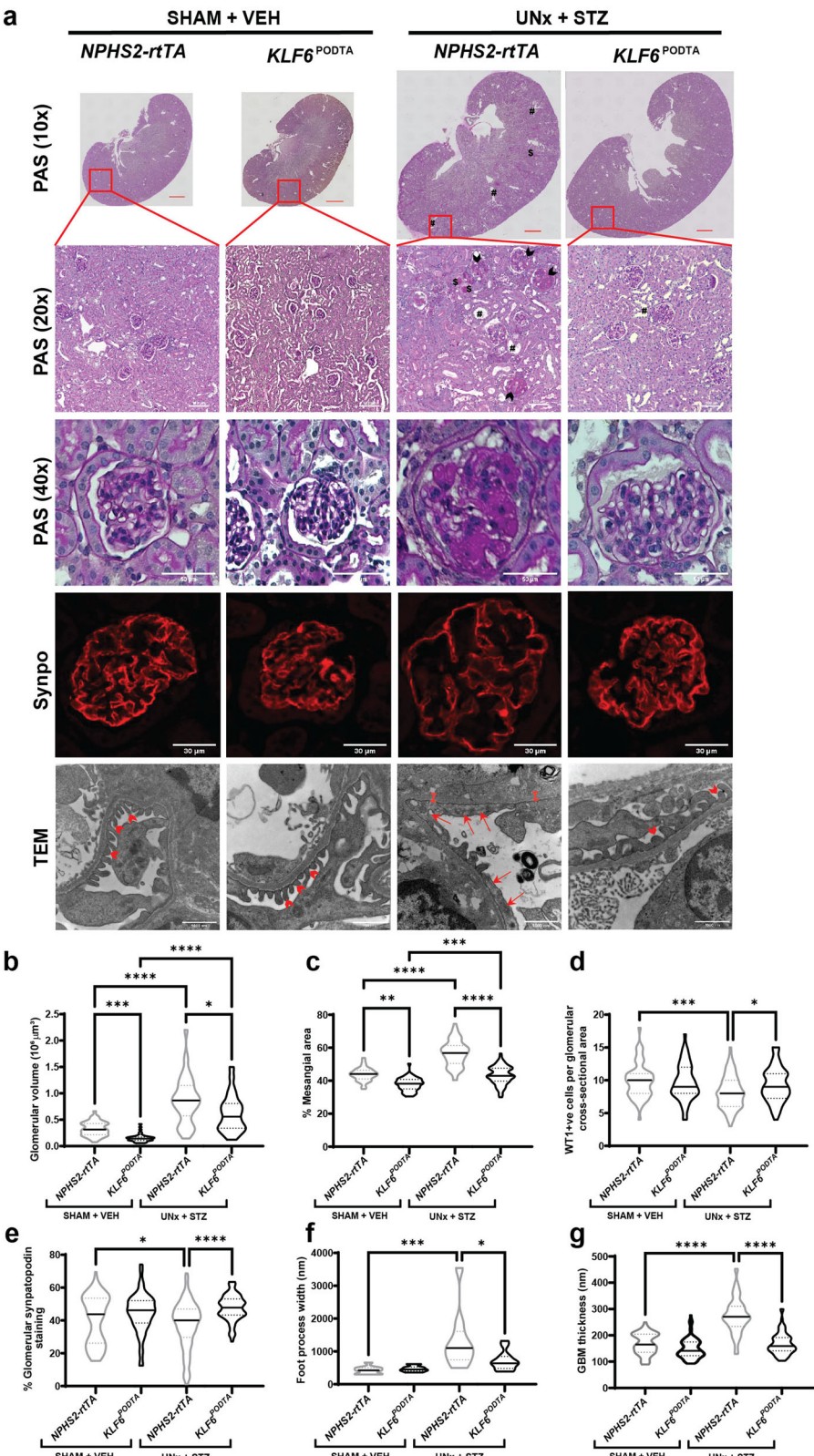

libraries for 9,894 nuclei that passed all quality control checks (Supplementary Fig. 6a–c). Dimensionality reduction was performed using R-packages *Signac 1.8.0*[20] and *Seurat 4.3.0*[21], and clusters were identified and assigned using anchors transferred from the snRNA-seq data. Label transfer was very successful in mapping the snATAC-seq data as indicated by the high prediction scores (Supplementary Fig. 6d). Nuclei from all 18 clusters in our snRNA-seq data were identified in the

snATAC-seq data (Fig. 4a). To compare conformity between the snRNA and snATAC data, we plotted gene activity metrics for the clusters using the top significant marker genes from the snRNA-seq data (Fig. 4b). Gene activity score predicts the level of gene expression based on the accessibility of regulatory elements in the vicinity of the gene[22]. Majority of the clusters showed exemplary correlations between gene expression of the selected marker gene and its

**Fig. 2 | *KLF6^PODTA* mice demonstrate a reduction in glomerular and tubular injury compared to *NPHS2-rtTA* mice under diabetic conditions.**
**a** Representative images of PAS staining at full scan (10x), low (20x) and high (40x) magnifications, synaptopodin staining, and transmission electron microscopy (TEM) of glomerular ultrastructure. Black arrowheads indicate sclerotic glomeruli, $ = protein casts, and # = tubular dilation. Top panel: red scale bar = 1000 μm. TEM: red arrows indicate effaced foot processes; red arrowheads point to healthy foot processes; red calipers show thickened glomerular basement membrane (GBM). Quantification of (**b**) glomerular volume; $p = 0.0001$ for SHAM + VEH (*NPHS2-rtTA* vs *KLF6^PODTA*), $p = < 0.0001$ for *NPHS2-rtTA* (SHAM + VEH vs UNx+STZ) and *KLF6^PODTA* (SHAM + VEH vs UNx+STZ), $p = 0.0228$ for UNx+STZ (*NPHS2-rtTA* vs *KLF6^PODTA*), (**c**) mesangial expansion; $p = 0.0001$ for SHAM + VEH (*NPHS2-rtTA* vs *KLF6^PODTA*), $p = <0.0001$ for *NPHS2-rtTA* (SHAM + VEH vs UNx + STZ) and *KLF6^PODTA* (SHAM + VEH vs UNx+STZ *KLF6^PODTA*), $p = 0.0228$ for UNx+STZ (*NPHS2-rtTA* vs *KLF6^PODTA*), and

(**d**) podocyte number (WT1^+ve Hoechst^+ve cells) per glomerular cross-section; $p = 0.0005$ for *NPHS2-rtTA* (SHAM + VEH vs *NPHS2-rtTA*), $p = 0.0428$ for UNx+STZ (*NPHS2-rtTA* vs *KLF6^PODTA*). ($n = 20$ glomeruli/mouse, $n = 3$–5 mice/group, $*p < 0.05$, $**p < 0.01$ $***p < 0.001$ $****p < 0.0001$ Kruskal–Wallis test with Dunn's post-test). **e** Quantification of glomerular synaptopodin staining ($n = 20$ glomeruli/mouse, $n = 3$–5 mice/group; $p = 0.0374$ for *NPHS2-rtTA* (SHAM + VEH vs UNx + STZ), $p = < 0.0001$ for UNx+STZ (*NPHS2-rtTA* vs *KLF6^PODTA*), $*p < 0.05$, $****p < 0.0001$; Mann–Whitney test, two-sided). **f** Quantification of podocyte foot process effacement measured as foot process width (nm); $p = 0.0001$ for *NPHS2-rtTA* (SHAM + VEH vs UNx+STZ), $p = 0.0262$ for UNx + STZ (*NPHS2-rtTA* vs *KLF6^PODTA*) and (**g**) GBM thickness (nm) ($n = 3$ mice/group; $p = <0.0001$ for *NPHS2-rtTA* (SHAM + VEH vs UNx + STZ), $p = <0.0001$ for (UNx + STZ *NPHS2-rtTA* vs *KLF6^PODTA*), $*p < 0.05$, $***p < 0.001$, $****p < 0.0001$; Kruskal–Wallis test with Dunn's post-test). Source data are provided as a Source Data file.

corresponding gene activity score (Fig. 4b). We also observed that all the clusters exhibited unique cell type-specific chromatin accessibility (Fig. 4c). To explore the potential chromatin accessibility changes induced by *KLF6* induction in the podocyte cluster, we performed differential accessibility analysis in the podocyte cluster between both groups (Fig. 4d, Supplementary Data 3). We also identified several chromatin organization genes such as Nuclear Receptor Coactivator 1 (*Ncoa1*), Nuclear Receptor Coactivator 2 (*Ncoa2*), Bromodomain Containing 1 (*Brd1*), Lysine Demethylase 5 C (*Kdm5c*), Lysine Demethylase 2 A (*Kdm2a*), PHD Finger Protein 20(*Phf20*), MSL Complex Subunit 2 (*Msl2*), GATA Zinc Finger Domain Containing 2B (*Gatad2b*) and, Polybromo 1 (Pbrm1) are differentially expressed in *KLF6^PODTA* podocytes (Supplementary Fig. 5f), suggesting that KLF6 might regulate chromatin reorganization. In addition, the *KLF6^PODTA* podocytes possessed chromatin regions that were more accessible compared to the *NPHS2-rtTA* podocytes (Fig. 4d). Motif enrichment analysis of the differentially more accessible chromatin regions in the *KLF6^PODTA* podocytes showed a high enrichment for several classes of transcription factors involved in podocyte differentiation such as WT1, KLF15, Transcription Factor 21 (TCF21), including KLF6, among others (Fig. 4e, Supplementary Data 4). Subsequent gene ontology (GO) enrichment analysis for the differential accessible chromatin regions showed an enrichment for biological processes involving the actin cytoskeleton such as cell-cell communication, Ca^2+ signaling, VEGF signaling, receptor for tyrosine kinase signaling, and syndecan interactions, with higher statistical significance in the *KLF6^PODTA* accessible chromatin regions (Fig. 4f). Collectively, these data suggest that podocyte-specific induction of *KLF6* preserves podocyte health by increasing the accessibility to podocyte pro-differentiation transcription factors.

**Podocyte-specific induction of *KLF6* induces CaMK1D signaling in the 1st segment of the proximal tubule**
Initial clustering of snRNA-seq data from all 4 groups demonstrated a transcriptionally distinct cluster resembling some, but not all components of the known segments of the PT, which we labeled "preconditioned-PT" (Fig. 3a). Interestingly, this cluster was predominant in both the nondiabetic and diabetic *KLF6^PODTA* groups, suggesting these transcriptional changes in this preconditioned-PT cluster are driven primarily by the induction of *KLF6* in the podocytes (Fig. 5a). In addition, the UMAP plots for all 4 groups showed a relative shift in cell population between the preconditioned-PT and the other PT groups (PTS1-S2, PTS1-S3, PTS3). To characterize this preconditioned-PT cluster and explore its functional significance, we initially determined the top DEGs for this cluster (Supplementary Data 2). Key DEGs in this cluster included calcium/calmodulin dependent protein kinase ID (*Camk1d*), Cms1 ribosomal small subunit homolog (*Cmss1*), aminoacylase 3 (*Acy3*), and glutathione peroxidase 3 (*Gpx3*) (Fig. 5b). We subsequently interrogated our preconditioned-PT cluster using previous reported markers for PT-S1-S3, repairing/Injured PT and failed repair PT cell markers[23–27] and found none of the repairing/injured or

"failed repair" cell markers are expressed by the preconditioned-PT cell cluster, suggesting that this cluster is distinct from previously reported post-injury PT clusters (Fig. 5c). Subsequent pathway enrichment analysis using WikiPathways[17] for these upregulated DEGs demonstrated an enrichment in metabolic pathways, such as electron transport chain, glycolysis & gluconeogenesis, amino acid metabolism, tricarboxylic acid cycle (TCA) cycle, and peroxisome proliferator-activated receptors (PPAR) signaling, tryptophan metabolism, and fatty acid metabolism, which are collectively pathways known to be enriched in the PT segments under basal conditions (Fig. 5d). Furthermore, among the upregulated DEGs in the preconditioned-PT cluster, *Camk1d* and *Cmss1* were significantly upregulated in the diabetic *KLF6^PODTA* as compared to the other groups (Fig. 5e). To explore the potential transcriptional paths between the tubular segments and the preconditioned-PT, we performed trajectory analysis on the snRNA sequencing data using monocle (Supplementary Fig. 7a). While the other PT segments, namely PT(S1-S2), PT(S1-S3), PT(S3) and PT(S3)/LH(DL), showed similar ordering across pseudotime constrained mainly on one side of branch point 1, the preconditioned-PT spanned across both sides of branch point 1, showing both its similarity as well as uniqueness from the other PT segments (Supplementary Fig. 7b). Expression of the key DEGs of the preconditioned-PT (*Camk1d*), PT(S1-S2) (*Slc5a12*), PT(S1-S3) (*Erc2*), PT(S3) (*Keg1*), PT(S3)/LH(DL) (*Cyp7b1*) and PEC/Proliferating PT (*Cd44*), localized similarly to their pseudotime ordering patterns (Supplementary Fig. 7d). We further interrogated the DEGs responsible for the branch-dependent expression across branch point 1 and the pathways they are involved in using clusterprofiler (Supplementary Fig. 7c, e). We found fatty acid beta-oxidation, glycolysis and gluconeogenesis, oxidative phosphorylation, and electron transport chain among the significant pathways (Supplementary Fig. 7e). These data suggest that a shift towards a highly metabolically active PT characterizes the transcriptome of the preconditioned-PT cluster.

To determine the spatial location of these unique PT cell populations, we costained for CaMK1D (high enrichment in the preconditioned-PT cluster) and lotus lectin in *NPHS2-rtTA* and *KLF6^PODTA* mice, which showed granular and apical staining specifically in the 1st segment of the PT, proximal to the podocytes (Fig. 5f, Supplementary Fig. 7f). Furthermore, this PT-specific CaMK1D expression was increased in the *KLF6^PODTA* mice as compared to *NPHS2-rtTA* mice (Fig. 5f, g). We also validated that *Camk1d* mRNA expression was enriched in PT cells in isolated primary PT cells as compared to neighboring parietal epithelial cells (PECs), and podocytes (Fig. 5h). In addition, CaMK1D was enriched in the PT segments with coexpression of CaMK1D in PHA-E^+ve cells in healthy donor nephrectomies (Fig. 5i). Interestingly, this PT-specific CaMK1D expression was reduced in kidney biopsies with early (<30% fibrosis), prior to significant loss of PT segments, as well as late (>30% fibrosis) DKD as compared to control specimens (Fig. 5i, j). Based on these data, the sole induction of podocyte-specific *KLF6* triggers CaMK1D signaling and pro-metabolic

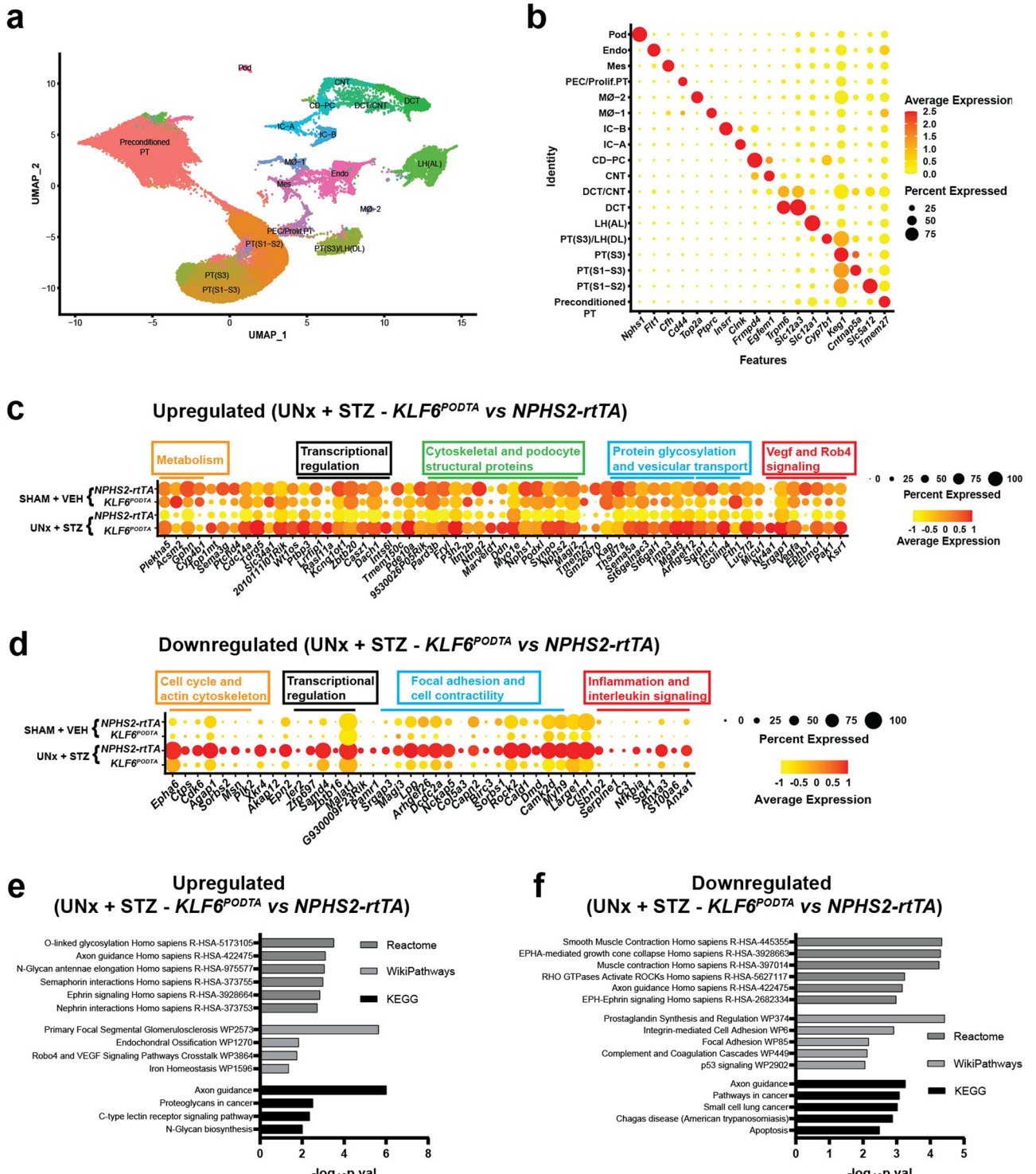

**Fig. 3 | snRNA-seq analysis showing the clustering signature and composition.**
**a** Uniform Manifold Approximation and Projection (UMAP) plot shows the 78,979 nuclei, mapping to 18 clusters. **b** Dot plot showing cluster identities aligned to canonical cell types in the adult mouse kidney based on a variety of cell type-specific marker genes. PT proximal tubules, LH(DL) Loop of Henle (Descending loop), LH(AL) Loop of Henle (Ascending loop), DCT distal convoluted tubule, CNT connecting tubule, CD-PC collecting duct principal cell, IC intercalated cell, Mø macrophage, PEC parietal epithelial cell, Prolif.PT Proliferating proximal tubules, Mes mesangial, Endo endothelial, Pod podocyte. Dot plot of all significantly (**c**) upregulated and (**d**) downregulated genes in the podocyte cluster between

$KLF6^{PODTA}$ vs $NPHS2$-$rtTA$ mice under diabetic conditions. Expression level of these genes is shown by the heat map and the percentage of cells expressing the gene in the cluster is shown by the size. The default statistical test (Wilcoxon Rank Sum test, adjusted $p$-value based on Bonferroni correction) in $Seurat$ package was used. Significantly expressed genes are determined by adjusted $p < 0.05$. Reactome, WikiPathways and KEGG enrichment analysis of these (**e**) upregulated and (**f**) downregulated genes in the podocyte cluster between $KLF6^{PODTA}$ vs $NPHS2$-$rtTA$ mice under diabetic conditions. The default statistical test (fisher's exact test) in $Enrichr$ was used.

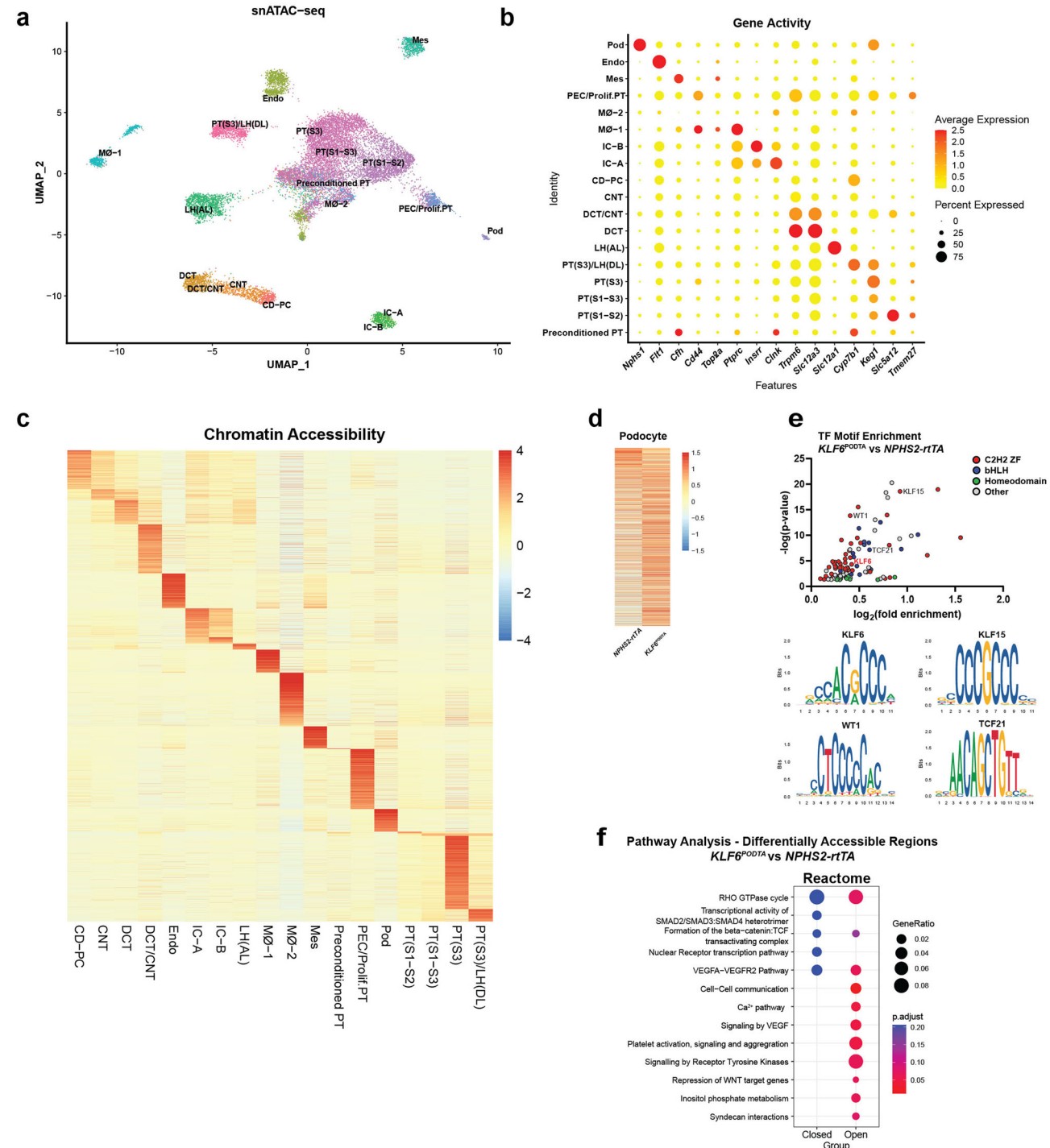

**Fig. 4 | snATAC-seq analysis showing the gene activity, chromatin accessibility and motif enrichment analysis at baseline. a** UMAP showing predicted annotation of the snATAC-seq data following integration and label transfer from the snRNA-seq data. **b** Dot plot showing gene activity aligned to canonical cell types in the adult mouse kidney based on a variety of cell type-specific marker genes. PT proximal tubules, LH(DL) Loop of Henle (Descending loop), LH(AL) Loop of Henle (Ascending loop), DCT distal convoluted tubule, CNT connecting tubule, CD-PC collecting duct principal cell, IC intercalated cell, Mø macrophage, PEC parietal epithelial cell, Prolif.PT Proliferating proximal tubules, Mes mesangial, Endo endothelial, Pod podocyte. **c** Heatmap showing of average number of Tn5 cut sites within a differentially accessible region (DAR) for all cell-types. **d** Heatmap showing

differential chromatin accessibility in the podocyte cluster of *KLF6*^PODTA vs *NPHS2-rtTA*. **e** TF motif enrichment analysis was carried out on genes with differential accessible chromatin in the podocyte cluster (*KLF6*^PODTA vs *NPHS2-rtTA*); genes with C2H2 zinc finger (C2H2 ZF) motifs are shown in red, basic helix−loop−helix (bHLH) in blue, Homeodomain in green and others are shown in gray. DNA motifs that are overrepresented in a set of peaks that are differentially accessible including KLF6, KLF15, WT1, and TCF21 are shown below. The default statistical test (fisher's exact test) in *Signac* package was used. **f** Reactome pathways associated with DARs in the podocytes are shown. Accessible terms in *KLF6*^PODTA are shown in red. The default statistical test (fisher's exact test) in *clusteProfiler* package was used.

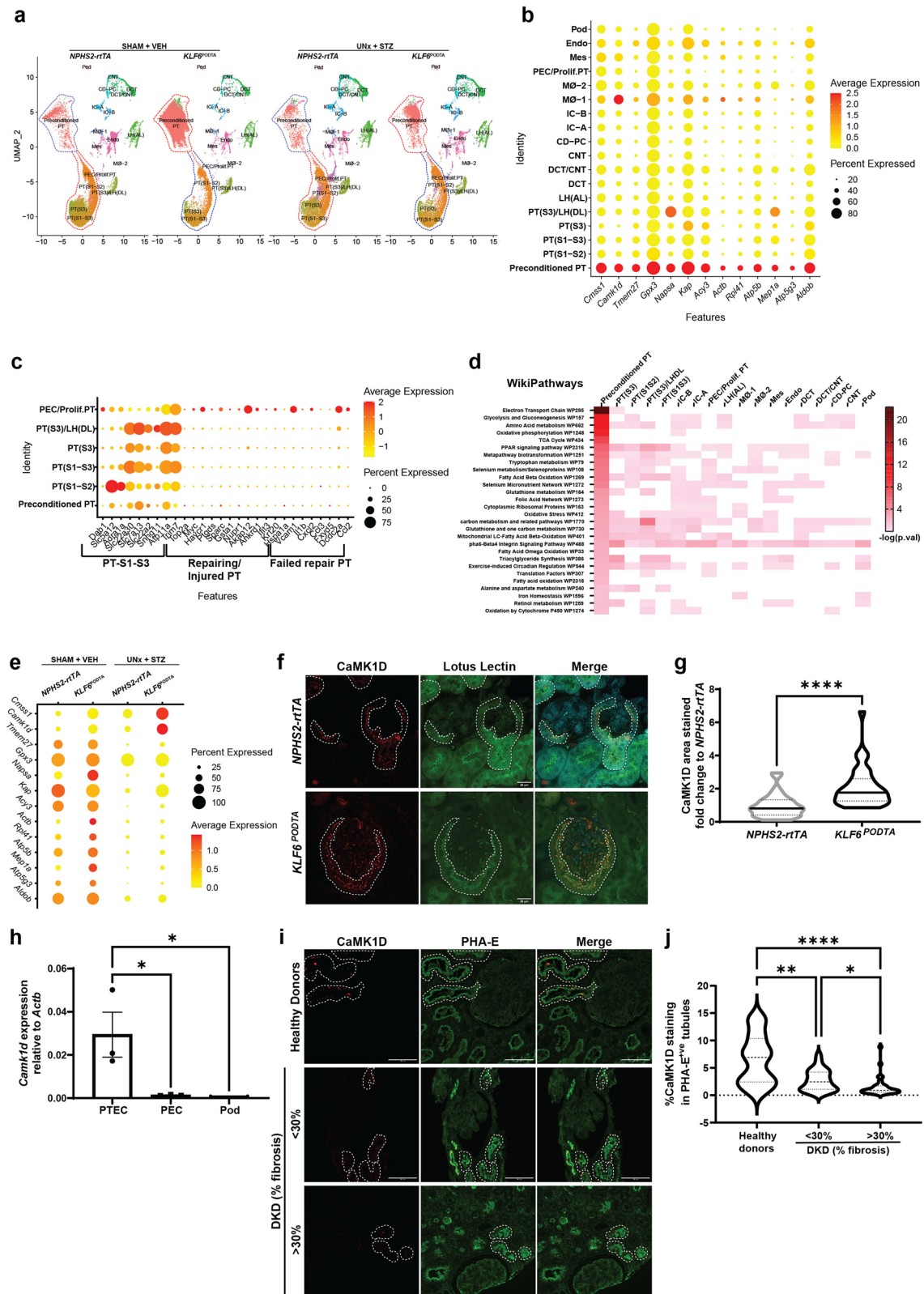

pathways in the 1st segment of PT, which might precondition the PT cells against diabetic injury.

## The requisite role of CaMK1D in the proximal tubule

In the setting of podocyte-specific *KLF6* induction, we report an increase in CaMK1D expression and an enrichment in DEGs involving oxidative phosphorylation and electron transport chain in the 1st

segment of PT, proximal to the podocytes. To ascertain the role of CaMK1D in the kidney, we initially knocked down *CAMK1D* in HK2 cells (*CAMK1D-shRNA*). *CAMK1D* knockdown was confirmed using qPCR, western blot and immunofluorescence staining (Fig. 6a–c). *CAMK1D-shRNA* cells exhibited a reduction in cell viability (cell count over time) as compared to *EV-shRNA* cells (Fig. 6d). These findings were validated with 3-(4,5-dimethylthiazol-2-yl)-2,5-diphenyltetrazolium bromide

**Fig. 5 | snRNA-seq analysis reveals a unique proximal tubule (preconditioned-PT) subpopulation predominant in _KLF6$^{PODTA}$_ mice. a** UMAP highlighting the relative shift in the nuclei between the preconditioned-PT and other PT clusters across the groups, red outline indicates higher nuclei count and blue outline indicates lower nuclei count. **b** Dot plot showing highly upregulated genes in the preconditioned PT cluster. **c** Dot plot showing injury markers from previously reported datasets (PT-S1-S3, repairing injured PT, failed repair PT) in the preconditioned-PT cluster as compared to other PT clusters. **d** Heatmap showing pathway enrichment analysis (WikiPathway) for differentially upregulated genes in the preconditioned PT cluster as compared to all other clusters. The default statistical test (fisher's exact test) in _Seurat_ package was used. **e** Dot plot showing differential expression of upregulated genes across the groups. **f** Representative images of CaMK1D and lotus lectin immunostaining. **g** Quantification of CaMK1D

staining per high power field (hpf - 20x) images ($n = 10$ hpf/mouse, $n = 3$ mice/group; $p = < 0.0001$, ****$p < 0.0001$; Mann–Whitney test). **h** _Camk1d_ expression (relative to mouse actin) in isolated primary proximal tubule cells (PTEC), parietal epithelial cells (PEC), and podocytes (Pod) ($n = 3$ mice/group, $p = 0.0381$ for PTEC vs PEC and $p = 0.0316$ for PTEC vs Pod, *$p < 0.05$; ordinary one-way ANOVA; data presented as mean ± SEM). **i** Representative images of CaMK1D and PHA-E staining in human biopsy tissue in healthy donor specimens, DKD (<30% and > 30% fibrosis) ($n = 3$ mice/group). **j** Quantification of CaMK1D staining in PHA-E$^+$ tubules ($n = 33–35$ tubules; $p = 0.0036$ for healthy donors vs < 30% fibrosis, $p = <0.0001$ for healthy control vs > 30% fibrosis, $p = 0.0284$ for < 30% fibrosis vs > 30% fibrosis, *$p < 0.05$, **$p < 0.01$, ****$p < 0.0001$; Kruskal–Wallis test with Dunn's post-test). Source data are provided as a Source Data file.

(MTT) assay (Fig. 6e). In addition, _CAMK1D-shRNA_ cells had reduced mitochondrial membrane potential as compared to _EV-shRNA_ cells (Fig. 6f). To assess for changes in mitochondrial structure, we initially stained for translocase of the outer mitochondrial membrane (TOM20), which demonstrated an increase in mitochondrial fragmentation in _CAMK1D-shRNA_ cells as compared to _EV-shRNA_ cells, as demonstrated by an increase in the % of cells with fragmented mitochondria as compared to tubular mitochondria (Fig. 6g, h). Since mitochondrial fission is inhibited with the phosphorylation of Dynamin related protein 1 (pDRP1) at Ser637[28], we observed that _CAMK1D-shRNA_ cells exhibited a reduction in pDRP1(ser637) expression as compared to _EV-shRNA_ cells (Fig. 6i).

To validate this detrimental effect of the loss of _CAMK1D_ on mitochondria specifically in primary PT cells, we pharmacologically inhibited CaMK1D with STO-609, which inhibits CAMKK, leading to a loss of phosphorylation and subsequent inactivation of CaMK1D[29,30]. Primary PT cells treated with STO-609 at 20 μg/ml and 50 μg/ml resulted in reduced cell viability (Fig. 6j) as well as a reduction in basal respiration, ATP production, maximal respiration, and spare respiratory capacity as compared to DMSO-treated cells (Fig. 6k, l). Furthermore, STO-609-treated cells also exhibited a reduction in basal respiration, ATP production, maximal respiration, and spare respiratory capacity as compared to DMSO-treated cells under normal glucose (NG, 5 mM) conditions, which was further exacerbated under high glucose (HG, 30 mM) conditions (Fig. 6m, n). TOM20 staining also showed an increase in mitochondrial fragmentation in STO-609-treated cells as compared to DMSO-treated cells under HG conditions (Fig. 6o, p). In addition, STO-609 reduced pDRP1(ser637) expression in primary PT cells as compared to DMSO under HG conditions, suggesting an increase in mitochondrial fission in the presence of pharmacological CaMK1D inhibition (Fig. 6q). Under diabetic conditions, we also observed that STO609 increased PT injury (loss of brush border staining, tubular dilatation, and reduced lotus lectin expression) as compared to DMSO treatment in the _KLF6$^{PODTA}$_ mice (Supplementary Fig. 8a, b). STO609-treated mice also had an increase in interstitial fibrosis (increase in picrosirius red and α-SMA expression) as compared to DMSO-treated mice, suggesting that CaMK1D inhibition exacerbated kidney injury under diabetic conditions (Supplementary Fig. 8a–c).

### Increased release of ApoJ from the KLF6$^+$ podocytes triggers PT CaMK1D signaling, leading to a reduction in PT injury

To ascertain the mechanism by which podocyte-specific _KLF6_ triggers CaMK1D signaling in PT cells, primary PT cells were initially treated with conditioned media from primary mouse podocytes isolated from _KLF6$^{PODTA}$_ mice as compared to _NPHS2-rtTA_ mice under NG and HG conditions. While the _NPHS2-rtTA_ podocyte secretome reduced cell respiration in PT cells, this was restored with exposure to the _KLF6$^{PODTA}$_ podocyte secretome under HG conditions (Fig. 7a, b). Tandem mass spectrometry was subsequently performed on the conditioned media, after removal of cell debris, to determine the differentially secreted

proteins between _KLF6$^{PODTA}$_ and _NPHS2-rtTA_ podocytes. A total of 838 proteins were identified, with 321 proteins significantly increased [fold change (FC > 1.2)] and 132 proteins significantly decreased (FC < 0.77) in the _KLF6$^{PODTA}$_ compared with the _NPHS2-rtTA_ conditioned medium (Fig. 7c, Supplementary Data 5). Key proteins such as ApoJ, calreticulin (CALR), clathrin light chain A (CLTA), clathrin light chain B (CLTB), collagen type IV alpha 1 (COL4A1) were differentially expressed in the conditioned medium from _KLF6$^{PODTA}$_ as compared to _NPHS2-rtTA_ podocytes. In comparison, a total of 142 proteins were identified, with 30 proteins significantly increased (FC > 1.3) in the urine proteome of _KLF6$^{PODTA}$_ as compared to _NPHS2-rtTA_ mice (Fig. 7d, Supplementary Data 5). Interestingly, ApoJ was uniquely enriched in both the podocyte secretome and the urine proteome of _KLF6$^{PODTA}$_ as compared to the _NPHS2-rtTA_ mice, suggesting an increase in the release of ApoJ from the podocytes in presence of _KLF6_ induction. ApoJ, also known as clusterin, is a glycoprotein that has been previously reported to have a salutary effect in the kidney[31–34]. Specifically, pre-treating podocytes with recombinant ApoJ has shown to be protective against apoptosis by reducing oxidative-stress induced under diabetic conditions[33]. To determine the expression of _ApoJ_ in the glomeruli under diabetic conditions, we interrogated previous reported microarrays using _Nephroseq_ to compare human biopsy specimens with DKD as compared to healthy donor nephrectomies[35,36]. _ApoJ_ expression was significantly increased in microdissected glomeruli in DKD patients compared to healthy donors and it inversely corelates with estimated GFR (Supplementary Fig. 9a, b). To ascertain potential ligand-receptor interactions between podocytes and PT cells, a previously validated database of known ligand-receptor interactions[37] was interrogated to predict potential ligands from the podocyte secretome and urine proteome as well as corresponding receptors from the DEGs in the preconditioned-PT cell cluster (Fig. 7e). ApoJ, which was identified in the podocyte secretome and the urine proteome of _KLF6$^{PODTA}$_ mice, demonstrated a potential ligand-receptor interaction with Lrp2 in the preconditioned-PT cluster (Fig. 7e). Interestingly, depending on the degree of glycosylation, the secretory form of ApoJ has been previously reported to serve as a molecular chaperone by binding to specific cell surface receptors to mediate its biological effects, such as endocytosis[38]. Immunohistochemistry confirmed that ApoJ was enriched in the podocytes and the apical portion of the 1st PT segment in _KLF6$^{PODTA}$_ as compared to _NPHS2-rtTA_ mice (Fig. 7f). Additionally, we costained ApoJ with podocyte marker, synaptopodin, and found increased colocalization of ApoJ in podocytes (Supplementary Fig. 9c). While ApoJ is present at high levels in the plasma and prevents complement deposition[39], we did not observe an increase in complement deposition (C3 and C5b-9 expression) in _KLF6$^{PODTA}$_ as compared to _NPHS2-rtTA_ mice (Supplementary Fig. 9d).

To determine whether the PT-protective effects of the _KLF6$^{PODTA}$_ podocyte secretome are driven largely by ApoJ, the conditioned media from _KLF6$^{PODTA}$_ and _NPHS2-rtTA_ podocytes were incubated with goat anti-ApoJ antibody (1:100) as compared to goat IgG antibody for 24 h prior to being administered to primary PT cells. While the _KLF6$^{PODTA}$_

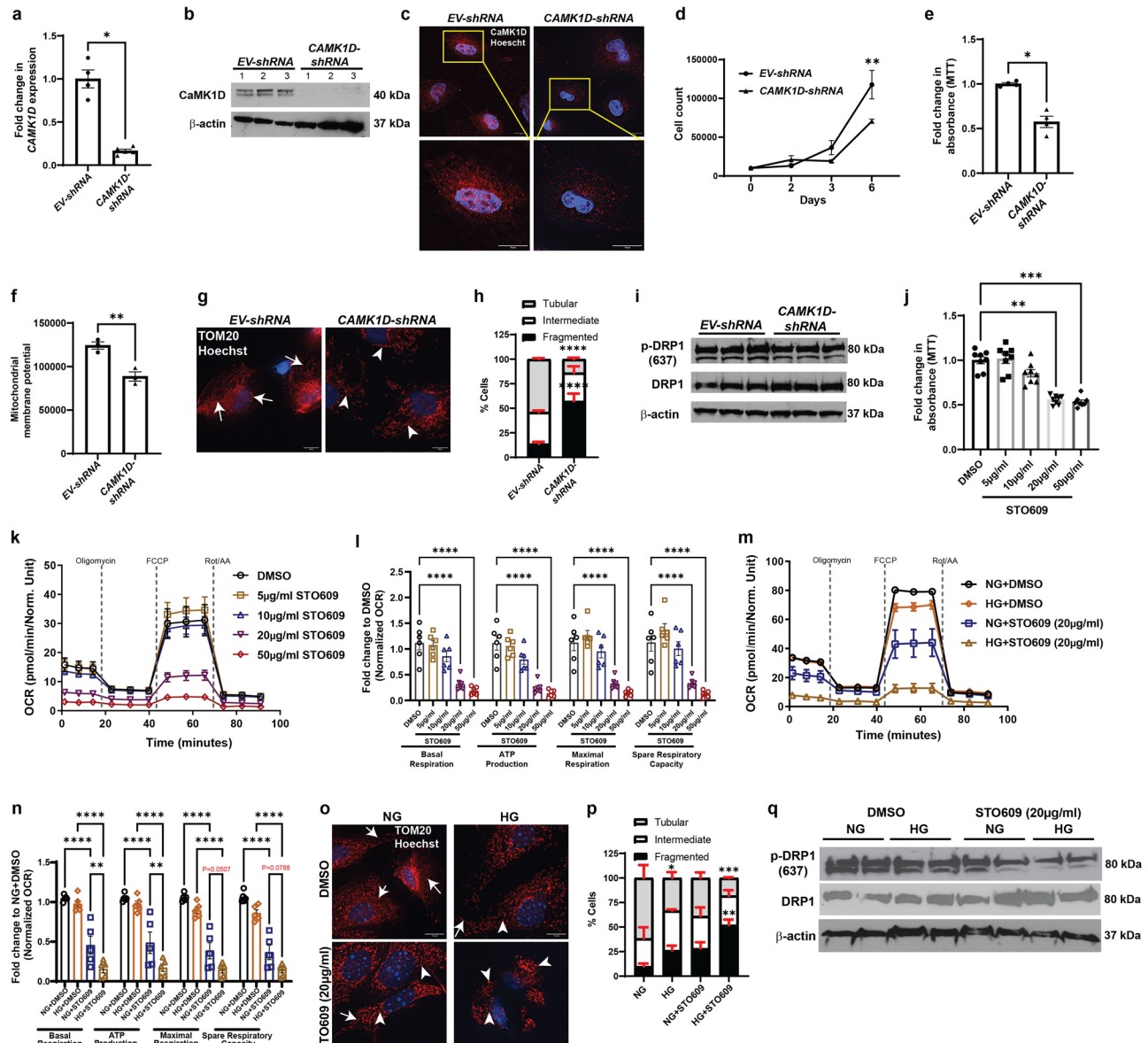

**Fig. 6 | Genetic and pharmacological inhibition of CaMK1D induces mitochondrial injury and reduces cell survival. a** Fold change in *CAMK1D* expression (n(biological replicates) = 4 for *EV-ShRNA* and n(biological replicates) = 5 for *CAMK1D-shRNA*, *p* = 0.0159, *\*p* < 0.05; Mann–Whitney test, two-sided; data presented as mean ± SEM. **b** Western blots for CaMK1D and β-actin. **c** Representative images for CaMK1D and Hoechst immunostaining. **d** Cell count at days 0, 2, 3 and 6 (n(biological replicates) = 3, *p* = 0.0025, *\*\*p* < 0.01; two-way ANOVA; data presented as mean ± SEM. **e** Cell viability (n(biological replicates) = 4, *p* = 0.0286, *\*p* < 0.05; Mann–Whitney test, two-sided; data presented as mean ± SEM. **f** Mitochondrial membrane potential (n(biological replicates) = 3, *p* = 0.0071, *\*\*p* < 0.01; Welch's *t*-test, two-sided; data presented as mean ± SEM. **g** Representative images for TOM20 and Hoechst staining. Arrow-tubular mitochondria; Arrowheads-indicates fragmented mitochondria. **h** % of cells with tubular, intermediate, and fragmented mitochondria (n(biological replicates) = 30, for three independent experiments, *p* = <0.0001 for tubular and fragmented mitochondria *\*\*\*\*p* < 0.0001, two-way ANOVA; data presented as mean ± SEM. **i** Western blots for pDRP1(637), DRP1, and β-actin. **j** Cell viability in 1° PT cells (n(biological replicates) = 8, *p* = 0.0039 for DMSO vs STO609 (20 μg/ml) and *p* = 0.0007 for DMSO vs STO609 (50 μg/ml),

*\*\*p* < 0.01, *\*\*\*p* < 0.001; Kruskal–Wallis test with Dunn's posttest; data presented as mean ± SEM. **k–l** OCR in 1° PT cells (n(biological replicates) = 6, *p* = < 0.0001 for DMSO vs STO609 (20 μg/ml and 50 μg/ml), *\*\*\*\*p* < 0.0001; one-way ANOVA; data presented as mean ± SEM. **m, n** OCR with quantifications in 1° PT cells in normal (NG) and high glucose (HG) conditions (n(biological replicates) = 5, *p* = 0.0086 for basal respiration (NG + STO609 vs HG + STO609), *p* = 0.0056 for ATP production (NG + STO609 vs HG + STO609), *p* = <0.0001 for NG + DMSO vs NG + STO609 and HG + DMSO vs HG + STO609 for all other significant comparisons, *\*\*p* < 0.01, *\*\*\*\*p* < 0.0001; one-way ANOVA; data presented as mean ± SEM.
**o, p** Representative images of TOM20 and Hoechst staining and quantification of % of cells with tubular, intermediate, and fragmented mitochondria in 1° PT cells Arrow-tubular mitochondria; Arrowheads indicates fragmented mitochondria (n(biological replicates) = 30, for three independent experiments, *p* = 0.0389 for tubular NG vs HG, *p* = 0.0014 for fragmented NG vs HG + STO609, *p* = 0.001 for tubular NG vs HG + STO609, *\*p* < 0.05, *\*\*p* < 0.01, *\*\*\*p* < 0.001; two-way ANOVA; data presented as mean ± SEM. **q** Western blots for pDRP(637), DRP1, and β-actin in 1° PT. Source data are provided as a Source Data file.

podocyte secretome improved PT basal respiration, maximal respiration, and spare respiratory capacity, blocking of ApoJ attenuated these protective effects on the mitochondria (Fig. 7g, h). To further validate these findings, we initially generated human podocytes with stable

*APOJ* overexpression (*Lenti-ORF–APOJ*) as compared to control podocytes (*Lenti-ORF-control*) (Supplementary Fig. 10a, b). PT cells treated with the conditioned media from *Lenti-ORF-APOJ* podocytes had higher basal respiration, ATP production, maximal respiration, and

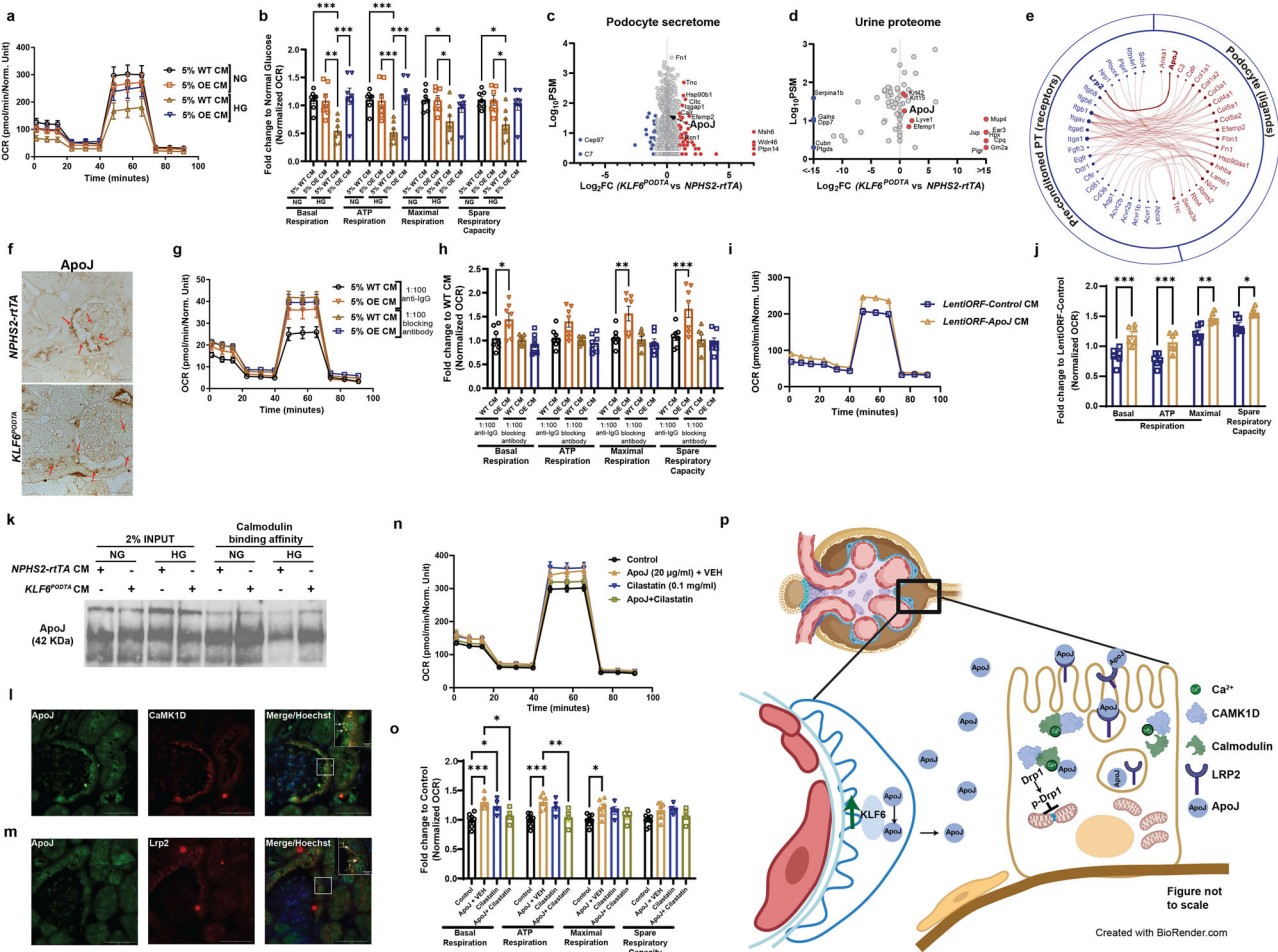

**Fig. 7 | Secretory ApoJ from *KLF6^PODTA* 1° mouse podocytes restores mitochondrial respiration under high glucose conditions. a, b** OCR and quantifications in 1° PT treated with conditioned media (CM) from 1° *KLF6^PODTA* (OE) versus *NPHS2-rtTA* (WT) podocytes in normal glucose (NG) and high glucose (HG). $n$ (biological replicates) = 7; basal respiration: $p = 0.0013$:OE CM (NG) vs WT CM (HG), $p = 0.0008$:OE CM (NG vs HG), $p = 0.0002$:WT CM vs OE CM (HG); ATP production: $p = 0.0006$:OE CM (NG) vs WT CM (HG), $p = 0.0004$:WT CM (NG vs HG), $p = <0.0001$:WT CM vs OE CM (HG); maximal respiration: $p = 0.0343$:WT CM (NG vs HG), $p = 0.0389$:OE CM (NG) vs WT CM (HG); spare respiratory capacity: $p = 0.0143$:OE CM (NG) vs WT CM (HG), $p = 0.0117$:WT CM (NG vs HG). **c, d** Volcano plot showing podocyte secretome and urine proteome from mice ($n = 3$). **e** Circos plot highlighting ligand-receptor interactions for upregulated proteins in podocyte secretome and DEGs in preconditioned-PT from snRNA-seq. **f** Representative images of ApoJ immunohistochemistry. Arrows-ApoJ staining ($n = 3$ mice/group, scale bar = 25 μm). **g, h** OCR and quantifications in 1° PT cells treated with CM + anti-ApoJ blocking antibody or CM + anti-IgG in HG ($n$ (biological replicates) = 7); WT CM vs OE CM: $p = 0.0259$(basal respiration), $p = 0.0026$(maximal respiration), $p = 0.0004$(spare respiratory capacity). **i, j** OCR and quantifications in 1° PT treated

with CM from differentiated human podocytes in HG ($n$ (biological replicates) = 6, $p = 0.0001$(basal respiration), $p = 0.0009$(ATP production), $p = 0.005$(maximal respiration), $p = 0.0156$(spare respiratory capacity)). **k** Western blot for immunoprecipitation of ApoJ in 1° PT treated with CM in NG/HG. **l, m** Representative images from co-immunostaining for ApoJ, CaMK1D, Lrp2, and Hoechst. Arrows-indicate ApoJ/CamK1D and ApoJ/Lrp2 colocalization. (scale bar = 50 μm, 5 μm for inset). **n, o** OCR and quantification in 1° PT treated with recombinant ApoJ ± cilastatin/VEH ($n$ (biological replicates) = 7 for control), ($n$ (biological replicates)=5) for ApoJ+VEH, (n(biological replicates) = 5) for ± cilastatin; basal respiration: $p = 0.0003$(control vs ApoJ + VEH), $p = 0.0157$(control vs cilastatin), $p = 0.0329$ (ApoJ + VEH vs ApoJ + cilastatin); ATP production:$p = 0.0001$ (control vs ApoJ + VEH), $p = 0.0086$ (ApoJ + VEH vs ApoJ + cilastatin); maximal respiration:$p = 0.0151$ (control vs ApoJ + VEH). **p** Proposed schematic of potential KLF6-ApoJ-CaMK1D signaling between podocytes and PT cells. For all data: *$p < 0.05$, **$p < 0.01$ ***$p < 0.001$, Kruskal–Wallis test with Dunn's posttest; data presented as mean ± SEM. Source data are provided as a Source Data file. **p** Created with BioRender.com released under a Creative Commons Attribution-NonCommercial-NoDerivs 4.0 International license https://creativecommons.org/licenses/by-nc-nd/4.0/deed.en.

---

spare respiratory capacity as compared to *Lenti-ORF-control* podocytes under HG conditions (Fig. 7i, j). Interestingly, the sole induction of *APOJ* also increased canonical podocyte markers of differentiated podocytes such as nephrin and synaptopodin, suggesting that ApoJ might also have salutary effects on podocyte health (Supplementary Fig. 10c).

Since calmodulin is critical component of CaMK1D activation and signaling, we tested the interaction between ApoJ and calmodulin by incubating the primary PT cell fractions that have been exposed to *KLF6^PODTA* and the *NPHS2-rtTA* podocyte secretome under NG and HG conditions using calmodulin-sepharose beads. While the ApoJ-calmodulin interaction was reduced in the PT cells when exposed to

*NPHS2-rtTA* podocyte secretome, this was restored with the *KLF6^PODTA* podocyte secretome under HG conditions (Fig. 7k). In addition, immunofluorescence staining demonstrated ApoJ colocalized with CaMK1D (Fig. 7l). Lrp2 has been reported previously as the endocytic receptor for ApoJ[40]. Since the molecular weight of Lrp2 limits co-immunoprecipitation studies, we conducted immunofluorescence staining with quantification using *CellProfiler* to demonstrate that ApoJ colocalizes with Lrp2 in the proximal tubules (Fig. 7m, Supplementary Fig. 11). We also treated the PT cells with recombinant ApoJ which led to an increase in PT basal respiration, ATP respiration, and maximal respiration. Finally, concurrent treatment with cilastatin, inhibitor of Lrp2 receptor activity[41–43], attenuated the salutary effects of ApoJ in

PT cells as demonstrated by a decrease in basal respiration, ATP-linked, and maximal respiration (Fig. 7n, o). Collectively, these data suggest that the kidney-protective effects of podocyte-specific KLF6 might, in part, be mediated through ApoJ-CaMK1D signaling in PT cells.

## Discussion

A large body of literature has focused on factors that drive kidney injury in diabetes, but little is known about mechanisms that confer resistance to progression of DKD. These unexplored mechanisms that delay progression of DKD might serve as potential targets for therapy in those individuals that progress rapidly. While podocyte injury contributing to glomerular dysfunction is an early indicator of kidney injury in diabetes, PT injury correlates with the decline in kidney function[3,44]. To date, mechanisms mediating podocyte to PT injury remain elusive in DKD. In this study, we demonstrate that the podocyte-specific induction of *KLF6* attenuates kidney injury in a murine model of diabetes. In addition to the salutary effects in the podocyte, the induction of podocyte-specific *KLF6* preconditions the PT from injury under diabetic conditions. By utilizing a combination of snRNA-seq, snATAC-seq, and LC-MS/MS, we report that podocyte-specific KLF6 triggers the release of ApoJ, which subsequently activates CaMK1D-mediated preservation of mitochondrial dynamics and function in the first segment of PT, proximal to podocytes (Fig. 7p). To date, this is the first study to demonstrate a novel mechanism by which the podocyte directly regulates PT mitochondrial health to attenuate the progression of DKD.

Our previous studies demonstrated the detrimental effects of podocyte-specific knockdown of *Klf6* in murine models of glomerular disease (i.e., Focal Segmental Glomerulosclerosis and DKD)[9,10]. Here, conversely, the induction of KLF6 specifically in podocytes ameliorated albuminuria and improved kidney function as well as histological features consistent with DKD. We also previously showed that induction of KLF6 in cultured podocytes attenuated detrimental effects of adriamycin in cultured podocytes[9]. However, similar to other tissues, this reno-protective effect of KLF6 appears to be cell-context dependent in the kidney. For instance, we recently reported the detrimental role of PT-specific KLF6 in a murine model of post-DNA damage in PT cells[45]. Furthermore, this contrasting cell-specific role of KLF6 is not restricted to the kidney, with opposing roles in the liver (hepatocytes versus hepatic stellate cells)[46] as well as in the heart (cardiac myocytes versus cardiac fibroblasts)[47]. One potential explanation in the kidney is that *Klf6* is expressed at a higher level in podocytes as compared to PT cells under basal conditions[48], thereby suggesting a requisite physiological role in the podocytes. Nonetheless, studies investigating the mechanisms regulating cell-specific contrasting roles of KLF6 will be important to understand its biology in kidney health and disease.

CaMKs (CaMK1, CaMK2, CaMK4, CaMKK) are multifunctional calcium/calmodulin-dependent protein kinases that have wide specificity but regulate several critical cellular functions in multiple cell types[49–52]. While CaMKs are well studied in the brain, their biology in the kidney is not well understood. A few studies have reported on the detrimental role of CaMK2 and CaMK4 activation in the kidney[53–56], but CaMK1D remains unexplored in the kidney. In genome-wide association studies (GWAS), single nucleotide polymorphisms in the *CAMK1D* loci are associated with increased risk of Type 2 Diabetes Mellitus[57–62]. Validation of our snRNA-seq with immunostaining demonstrates an enrichment of CaMK1D expression in the first segment of PT, proximal to podocytes. Interestingly, genetic and pharmacological inhibition of CaMK1D reduced pDRP1(S637) expression, leading to an increase in mitochondrial fragmentation with a reduction in mitochondrial membrane potential and respiration. DRP1 is a critical regulator of mitochondrial dynamics and posttranslational modifications are critical to its function[63]. Previous studies report that CaMKI might be involved in the regulation of DRP1 activity[64], but the mechanism(s) mediating this process remains to be investigated. Furthermore, future

studies are needed to examine the interplay between CaMK1D and other isoforms of CaMKI signaling on pDRP1 activity and mitochondrial health[64–66]. Nonetheless, this is the first study, to date, to demonstrate the protective role of CaMK1D in the kidney.

Another key finding in this study is the role of ApoJ in mediating the salutary effects of podocyte-specific KLF6 on CaMK1D activation in the PT. ApoJ undergoes posttranslational modification, largely glycosylation, in the endoplasmic reticulum and golgi apparatus prior to being released extracellular space. Secretory ApoJ serves as a molecular chaperone to regulate various cellular processes such as lipid transport, cell differentiation, membrane cycling, apoptosis, and cell-cell interactions[67]. Several studies have implicated ApoJ in a host of human diseases[68–72], ranging from myocardial injury[73] to post-ischemic brain injury[74] to Alzheimer's disease[75]. In the kidney, ApoJ expression has been previously shown to increase post-injury[76–79]. Specifically, both protein and mRNA expression levels of ApoJ are increased in podocytes in diabetic mice and in kidney biopsies with early DKD[33]. Similarly, we validated this in expression arrays deposited in *Nephroseq*. In addition, ApoJ levels in the urine have been associated with increased tubular damage in patients with diabetes[80]. We also observed an enrichment in ApoJ levels in both the urine and podocyte secretome from *KLF6^PODTA* as compared to *NPHS2-rtTA* mice by LC-MS/MS. Interestingly, this increase in levels of secretory ApoJ might be in response to podocyte injury, since we observed that blocking ApoJ in the *KLF6^PODTA* podocyte secretome attenuated the salutary effects in the PT cells under HG conditions. Furthermore, *ApoJ* knockout mice develop glomerulopathy with aging[76] and individuals with glomerular disease demonstrate an overall depletion in the pool of ApoJ with progressive disease[32]. In addition, salutary effects of ApoJ have been shown in other models of kidney injury[81,82].

However, to date, the mechanism by which secretory ApoJ attenuates kidney injury remains poorly understood[83]. Here, we demonstrate that induction of KLF6 triggers the release of secretory ApoJ, which undergoes cellular uptake by Lrp2, triggering the activation of CaMK1D by binding to calmodulin in PT cells. snRNA-seq analysis demonstrated an enrichment in pathways associated with N-glycosylation in the DEGs with KLF6 binding sites. Glycosylated intracellular ApoJ forms may act as a redox sensor under oxidative stress conditions and are essential for its chaperone activity as well as release of ApoJ into the extracellular space[84,85]. Therefore, the induction of KLF6 could potentially increase glycosylation of ApoJ, thereby triggering the release of secretory ApoJ from the podocytes. However, additional studies are required to test the mechanism by which KLF6 leads to glycosylation of ApoJ. Interestingly, ApoJ has been previously reported to serve as a ligand for Lrp2-facilitated endocytosis in the brain[40,86,87] and has a putative calmodulin binding domain containing three motifs (two 1–12, one 1–14)[88]. A recent study using Ingenuity Pathway Analysis also showed KLF6 motifs in the regulatory regions of *ApoJ* as well as potential interactions between calmodulin and Lrp2[89]. Calcium and calmodulin also play a role in supporting the endocytosis of ApoJ as calmodulin has been previously reported to serve as a calcium sensor for endocytosis in synapses[90]. Furthermore, the proximity of podocyte secretory ApoJ to Lrp2 and calmodulin-CaMK1D in the first segment of the PT, in combination with the direction of glomerular filtrate, could enhance the feasibility of this interaction. ApoJ has also been reported to protect endothelial cells by suppressing mitochondrial fission under diabetic conditions[91]. In addition, ApoJ has been reported to facilitate mitochondrial respiration in the healthy brain[92] and overexpression of *ApoJ* attenuated Drp1 activation, thereby inhibiting mitochondrial fission[93]. Based on our findings, we postulate that ApoJ-mediated activation of CaMK1D-pDRP1 signaling protects the PT against injury by inhibiting mitochondrial fission under diabetic conditions.

Collectively, these studies uncover a previously unreported mechanism by which the podocyte preconditions the proximal

tubule to attenuate mitochondrial injury and subsequent deterioration of kidney function under diabetic conditions. In addition to this requisite role of podocyte-specific KLF6 in podocytes, we also provide evidence for a new downstream signaling pathway involving secretory ApoJ-Lrp2-CaMK1D-pDrp1 between podocytes and the first segment of PT that might be critical for kidney health. Finally, maintaining secretory ApoJ from podocytes or enhancing CaMK1D signaling in the PT might be a key therapeutic strategy in attenuating the progression of DKD.

## Methods

### Generation and validation of $KLF6^{PODTA}$ mice

All animal studies were approved by the Stony Brook Animal Care and Use Committee and carried out in accordance with the National Institutes of Health standards. To generate $KLF6^{PODTA}$ mice, *NPHS2-rtTA* mice (FVB/N-Tg(NPHS2-rtTA2*M2)1Jbk/J, The Jackson Laboratory) were bred with the *TRE-KLF6* mice to generate mice with both transgenes on the *FVB/N* background. The *TRE-KLF6* transgene contained the (TetO)7/CMV regulatory element driving the full-length human KLF6 coding sequence (ORF021674) followed by the polyadenylation signal[45]. Transgene was purified from plasmid vector sequences and microinjected into the pronucleus of *FVB/N* single-celled embryos to generate *TRE-KLF6* mice. DNA extraction by Extracta DNA prep (Quanta Biosciences) from tails at 2 weeks of age and PCR to confirm the genotype. Primers for genotyping are provided in Supplementary Table 3.

To induce transgene expression, $KLF6^{PODTA}$ mice were fed TestDiet Modified Rodent Diet 5001 containing 0.15% doxycycline (DOX; El-Mel). $KLF6^{PODTA}$ mice and littermate controls remained on DOX continuously starting at 8 weeks of age. To validate induction of podocyte-specific *KLF6* in $KLF6^{PODTA}$ mice, we generated fluorescent labeled *TRE-GFP* mice and bred them with $KLF6^{PODTA}$ mice or *NPHS2-rtTA* mice. All mice were treated with DOX. Glomeruli were subsequently isolated from these mice using iron oxide and the glomerular and tubular fraction was separated using a magnet after digestion with collagenase for 45 min at 37 °C as previously described[94]. The purity of glomeruli was verified under microscopy. Both fractions were digested using collagenase, DNase and trypsin to make a single cell suspension. The single cell suspension for the glomerular fractions was sorted using fluorescence-activated cell sorting (FACS) to isolate GFP(+ve) cells. GFP(-ve) glomerular and tubular cells were also collected for further analysis. Podocyte-specific *KLF6* expression was determined in GFP(+ve) glomerular cells as compared to GFP(-ve) glomerular and tubular cells.

### Animal experiments

The animals in this study were housed in our animal facility with free access to chow (Lab diet 5053/Dox diet) and water and 12 h day/light cycle, in ambient temperature and humidity conditions.

Baseline urine was collected from the *NPHS2-rtTA* and $KLF6^{PODTA}$ mice. Mice were anesthetized and UNx was performed as previously described[95]. In brief, the blood vessels of the left kidney were ligated, and the kidney removed surgically, and the mice were monitored for a week after surgery. To induce diabetes, mice were administered STZ (50 mg/kg) in 50 mmol/L sodium citrate buffer (pH 4.5) by intraperitoneal injection over the course of 5 days[10,96]. On day 14 after the STZ injections, blood glucose was measured from the tail vein using a OneTouch glucometer (LifeScan)[94]. Diabetes was defined as sustained fasting blood glucose level > 250 mg/dL. All mice were euthanized using intraperitoneal injection of Ketamine/Xylazine (150/20 mg/kg) at 20 weeks of age.

For the STO-609 experiments, UNx-STZ $KLF6^{PODTA}$ mice were treated with DMSO or STO-609 (30 μm/kg) daily intraperitoneal injection for 2 weeks. The mice were euthanized using intraperitoneal injection of Ketamine/Xylazine (150/20 mg/kg) at 16 weeks of age.

### Measurement of albuminuria, serum urea nitrogen and creatinine

The animals were housed in a single mouse metabolic cage (Tecniplast) with free access to food and water, and urine was collected after 24 h. Albumin concentration was measured using ELISA assay kit (Bethyl Laboratory). Twenty-hour urine albumin concentration was calculated by multiplying total volume of urine collected and albumin concentration. Serum urea nitrogen levels were measured by a colorimetric detection method (Arbor Assay) according to the manufacturer's protocol. Serum creatinine levels were measured using the isotope dilution liquid chromatography-tandem mass spectrometer at the University of Alabama at the Birmingham O'Brien Core Center.

### Real-time PCR

Total RNA was extracted using TRIzol (Gibco) for cells and RNA easy kit (Qiagen) for kidney tissue. First-strand cDNA was prepared from total RNA using the SuperScript III First-Strand Synthesis Kit (Life Technologies), and cDNA was amplified using SYBR GreenER qPCR Supermix on ABI QuantStudio 3 (Applied Biosystems). Primers were designed using National Center for Biotechnology Information Primer-BLAST and validated for efficiency before application. Primer sequences are listed in supplemental information (Supplementary Table 4). Light cycler analysis software was used to determine crossing points using the second derivative method. Data was normalized to the housekeeping gene (*Actb* or *ACTB*) and presented as a fold increase compared to the control group using the $2^{-\Delta\Delta CT}$ method.

### Histopathology and morphometric studies by bright-field light microscopy

Mice were perfused with phosphate buffer saline (PBS) and the kidneys were fixed in 10% phosphate buffered formalin overnight and switched to 70% ethanol prior to processing for histology. Kidney tissue was embedded in paraffin by histology core facility at Stony Brook University and 4 μm thick sections were stained with periodic acid-Schiff (PAS) (Sigma-Aldrich), hematoxylin and eosin (H&E), and picrosirius red, according to previously reported protocols, and mounted in permanent mounting medium[45,94].

Mesangial expansion, and glomerular volume were quantified as previously described[10,94]. In brief, images were scanned, and glomerular areas were traced using ImageJ. Mean glomerular tuft volume (GV) was determined from mean glomerular cross-sectional area (GA) by light microscopy. GA was calculated based on average area of 20 glomeruli in each group and GV was calculated based on the following equation: $GV = \frac{\beta}{\kappa} \times GA^{3/2}$ [$\beta = 1.38$, the shape coefficient of spheres (the idealized shape of glomeruli), and $\kappa = 1.1$, the size distribution coefficient]. Mesangial expansion was defined as the PAS-positive and nuclei-free area in the mesangium. Quantification of mesangial expansion was based on 20 glomeruli cut at the vascular pole in each group.

Histological scoring for sclerotic glomeruli, tubular injury, interstitial fibrosis, and inflammation was performed in a blinded fashion using a semiquantitative scale from 0 to 3 (0 indicates none); 1 = mild (≤25%), 2 = moderate (>25%–50%) and 3 = severe (>50%) by the kidney histopathologist (M.P.R.).

### Immunofluorescence staining and immunohistochemistry

All kidney sections from mice were prepared for immunofluorescence staining in identical fashion as previously described[94]. Immunofluorescence staining was performed using mouse anti-WT1 (Santa Cruz, sc-7385, 1:50 dilution), goat anti-synaptopodin (Santa Cruz, sc21537, 1:200 dilution), rabbit anti-KLF6 (Santa Cruz, AP6588B, 1:150 dilution), mouse anti-α-SMA (Sigma-Aldrich, A1978, 1:10,000 dilution), rabbit anti-CaMK1D (Invitrogen, PA5-21957, 1:100 dilution), goat anti-ApoJ (Novus Biologicals, NBP1-06027, 1:100 dilution), mouse anti-Lrp2 (Novus Biologicals, NB110-96417, 1:100 dilution) and rabbit anti-

TOM20 (Abcam, ab78547, 1:100 dilution), Fluorescein-conjugated goat IgG to mouse complement C3 (MP Biomedicals, 085500, 1:100 dilution) and mouse anti-C5b-9 (Santa Cruz, sc-66190, 1:100 dilution) antibodies. After washing, sections were incubated with a fluorophore-linked secondary antibody (Alexa Fluor 647 Donkey anti-mouse, Fluor 488 Goat anti-rabbit, or Fluor 568 Donkey anti-rabbit from Life Technologies, 1:300 dilution). After counterstaining with fluorescein-labeled lotus lectin (Vector Labs, 1:100 dilution) and/or Hoechst (Invitrogen, 1:1000 dilution), slides were mounted in ProLong gold antifade mounting media (Invitrogen) and photographed under a Nikon Eclipse i90 microscope and DS-Qi1Mc camera.

Immunohistochemistry was conducted for ApoJ as previously described[9] using goat anti-ApoJ antibody (R&D, AF2747, 1:100 dilution). Briefly, slides were dewaxed, followed by rehydration with decreasing concentrations of ethanol (100%, 90%, 70%). After antigen retrieval using sodium citrate buffer at 120 °C, the endogenous peroxidases were blocked using hydrogen peroxide (3%) in methanol for 20 min, followed by blocking with 2% non-fat milk. The sections were incubated overnight with primary antibodies at 4 °C followed by anti-goat horseradish peroxidase secondary (Sigma Aldrich, AP200P, 1:300 dilution). The slides were then incubated with diaminobenzidine solution (Betazoid DAB Chromogen Kit, Biocare Medical) for 5 min, slides were then dehydrated in ethanol and xylene followed by mounting with permanent mounting media.

De-identified human biopsy specimens from University of Utah were categorized into early-stage (<30%) and late-stage (>30%) chronic tubulointerstitial fibrosis by a renal pathologist (M.P.R.). Control kidney biopsy specimens were acquired from the unaffected pole of kidneys that were removed because of renal cell carcinoma. The study was approved by the Stony Brook University Institutional Review Board (#798611).

Podocyte number was determined by dividing the number of WT1(+ve) Hoechst(+ve) podocytes per glomerular cross-sectional area. Glomerular synaptopodin expression was determined by measuring the percent area stained for synaptopodin in the glomerular cross-sectional area. Quantifications of lotus lectin and α-SMA in the cortex were determined by measuring the percentage area stained per cross-sectional area of a high-power field. All quantification was conducted using 20X high-power digitized images in ImageJ.

Immunofluorescence staining and quantification of mitochondrial fragmentation were done using TOM20 staining in cultured cells as previously described[9,97]. Briefly, mitochondrial morphology was categorized in each cell, by an investigator blinded to the experimental conditions, as tubular (>75% of mitochondria with tubular length > 5 mm), intermediate (25%–75% of mitochondria with tubular length > 5 mm), or fragmented (<25% of mitochondria with tubular length > 5 mm).

Quantification of colocalization of ApoJ with Lrp2 was done using *CellProfiler*[98]. The colocalization pipeline with minor modifications was utilized to measure the colocalization between fluorescently labeled ApoJ and Lrp2 to measure the degree of overlap between them. Briefly, the images were split into two channels (ApoJ and Lrp2), followed by illumination correction for both channels. The images were aligned to ensure accurate positioning of the features in both images (Supplementary Fig. 11b), and primary objects were identified for each channel along with the outliers as shown in Supplementary Fig. 11c, d. The primary objects from ApoJ channel that colocalize with the Lrp2 channel were measured (Supplementary Fig. 11e). A mask image indicating the colocalized areas was generated and the area occupied by the colocalized objects and Lrp2 was used for quantifications.

## Histopathology by transmission electron microscopy
Mice were perfused with PBS and then immediately fixed in 2.5% glutaraldehyde for transmission electron microscopy (TEM). Sections were mounted on a copper grid and photographed under a FEI BioTwinG2 transmission electron microscope. Briefly, negatives were digitized, and images with a final magnitude of ~ 10,000X were obtained[9]. The quantification of podocyte effacement and glomerular basement membrane (GBM) thickness was performed as previously described[96,99]. ImageJ was used to measure the length of the peripheral GBM and the number of slit pores overlying this GBM length was counted. The arithmetic mean of the foot process width (WFP) was calculated using the following: $WFP = \frac{\pi}{4} \times \frac{\sum GBM\ Length}{\sum slits}$; where $\sum$GBM length indicates the total GBM length measured in one glomerulus, $\sum$ slits indicates the total number of slits counted, and $= \frac{\pi}{4}$ is the correction factor for the random orientation by which the foot processes were sectioned[99]. Quantification of GBM thickness was performed as described[100,101]. The thicknesses of multiple capillaries were measured in 3–4 glomeruli per mouse. A mean of 120 measurements was taken per mouse (from podocyte to endothelial cell membrane) at random sites where GBM was displayed in the best cross section.

## Cell culture
1° mouse podocytes were isolated from GFP-labeled *KLF6PODTA* and *NPHS2-rtTA* male mice using FACS and were maintained on DOX (1 μg/ml) throughout the experimental conditions. Serum free media was collected from these podocytes to carry out proteomic analysis.

Mouse 1° PT cells were isolated from the kidney cortex of male mice after cardiovascular perfusion with PBS as previously described[102]. In brief, the cortex is digested using collagenase A at 37 °C for 1 h. The digested tissue is filtered, followed by multiple washing and resuspension in complete PT cell media (Dulbecco's modified Eagle's medium:F12 with 10 ng/L epidermal growth factor, 5 pM T3, 3.5 mg/L ascorbic acid, 25 μg/L prostaglandin E1, 25 μg/L hydrocortisone, 1 × insulin transferrin selenium supplement, 100 units/ml penicillin, and 100 μg/ml streptomycin).

Human kidney (HK2, ATCC, CRL-2190) cells with *CAMK1D* knockdown were generated using the Genecopoeia lentiviral shRNA system with the following construct, MSH079040-LVRU6GP-c, GGTGCTGTATATAAGAATCTT. In brief, lentiviral particles were produced by transfecting HEK 293 T cells with a combination of lentiviral expression plasmid DNA, pCD/NL-BH ΔΔΔ packaging plasmid, and VSV-G–encoding pLTR-G plasmid. For infection, viral supernatants were supplemented with 8 μg/ml polybrene and incubated with cells for 24 h. Cells expressing shRNA were selected with puromycin for 2–3 weeks before use in all studies. Real-time PCR, western blot, and immunofluorescence staining were performed to confirm *CAMK1D* knockdown.

Mouse 1° PT cells were treated with DMSO (vehicle) or CaMK1D inhibitor, STO-609 (5, 10, 20 and 50 μg/ml), for 24 h under normal glucose conditions. After dose optimization, the 1° PT cells were treated with 20 μg/ml of STO-609 for 24 h under normal (5 mM) and high glucose (30 mM) conditions. For measuring the effect of conditioned media on the 1° PT cells, the cells were treated with 5% conditioned media from 1° podocytes from *KLF6PODTA* and *NPHS2-rtTA* mice under normal glucose (NG) and high glucose (HG) conditions for 24 h. For the blocking experiments, the conditioned media (CM) was incubated with goat anti-clusterin antibody (R&D, AF2747) or goat IgG-antibody (1:100) for 24 h at 4 °C. The CM containing the blocking antibody was used to treat the 1° PT cells under NG and HG conditions.

The *LentiORF-APOJ* clone was purchased from Genecopoeia, and stable *APOJ* overexpression was achieved by lentiviral delivery. Cells expressing *LentiORF-APOJ* were selected with puromycin for 2–3 weeks prior to use in all studies. *LentiORF-control* serves as the GFP control vector. qPCR and western was performed to confirm *APOJ* overexpression.

Mouse 1° PT cells were treated with recombinant mouse ApoJ-His-tag (His-ApoJ) (20 μg/ml) (SinoBiological, 50485-M08H) and Cilastatin

(0.1 mg/ml) (MedChemExpress, HY-A0166A), for 24 h under NG conditions.

Oxygen consumption rate (OCR) and extracellular acidification rate (ECAR) were measured using a Seahorse XFe96 Analyzer (Agilent) in the presence or absence of mitochondrial function inhibitors as previously reported for all the different experimental conditions[45]. In brief, cells were seeded into 96 well plates at $2 \times 10^6$ cells per well, and 24 h later, the growth media was removed, cells were washed with PBS. DMSO, STO-609 or conditioned media was added with or without NG or HG, and 24 h later the media was removed and replaced with serum-free Seahorse DMEM, pH 7.4, supplemented with 1 mM glutamine and 2 mM pyruvate, NG or HG. After incubation in a $CO_2$ free incubator for 45–60 min, OCR and ECAR were measured at baseline or after acute injections of 1.5 μM oligomycin (Agilent), 3 μM carbonyl cyanide-p-(trifluoromethoxy) phenylhydrazone (FCCP) (Agilent), and 0.5 μM rotenone/antimycin A (Agilent) using the Seahorse XF Cell Mito Stress Test Kit (Agilent).

Mitochondrial membrane potential was measured using MitoProbe DiIC1(5) Assay Kit (Invitrogen). In brief, cells were trypsinized, washed with PBS, and incubated with 1,1′,3,3,3′,3′-hexamethylindodicarbo-cyanine iodide (DiIC1) alone or with DiIC1 and carbonyl cyanide 3- chlorophenylhydrazone (CCCP) and the difference in fluorescence intensity was measured between the groups using FACS.

Cell numbers were determined using the Countess 3 cell counter at days 0, 2, 3 and 6. Cell proliferation was also measured using the 3-(4, 5-dimethylthiazolyl-2)-2, 5-diphenyltetrazolium bromide (MTT) assay.

GFP-labeled primary mouse podocytes were obtained using FACS sorting. Other cell lines were validated in previous publications or by manufacturer's website

## Western blot assay
Protein lysates were collected from HK2 cells and 1° PT cells with a buffer containing 4% SDS, and protease phosphatase inhibitor. Protein lysates were subjected to immunoblot analysis using rabbit anti-CaMK1D (Proteintech, 13613-1-AP, 1:1000 dilution), rabbit anti-DRP1 (Invitrogen, MA5-26255, 1:1000 dilution), rabbit anti-phospho-DRP1(Ser637) (Invitrogen, PA5-101038, 1:1000 dilution), goat anti-ApoJ (R&D, AF2747, 1:1000 dilution), rabbit anti-ApoJ (Cell Signaling technology, 42143, 1:1000 dilution), and mouse anti-β-actin (Sigma-Aldrich, A1978, 1:5000 dilution) antibodies. Uncropped and unprocessed scans are included in the Source data file.

## Immunoprecipitation
Protein lysates were collected from 1° PT cells treated with 5% conditioned media with Pierce IP Lysis Buffer (Thermo Fisher Scientific, 87788) and Halt protease and phosphatase inhibitor cocktail (Thermo Fisher Scientific, 78438). Pull-down was performed using calmodulin (CaM)-sepharose beads (Abcam, ab286869) as previously described[103]. Briefly, protein lysates from 1° PT cells were incubated with CaM-sepharose beads at 4 °C for 3 h on a shaker, centrifuged, unbound protein was removed, and the beads were incubated with elution buffer at room temperature for 30 min on a shaker. The beads were then centrifuged, and the bound protein was subjected to western blot analysis to determine the amount of ApoJ pulled down with CaM beads.

## Single-nuclei isolation, sequencing, data processing, and analysis
Nuclei were isolated from the mouse kidney cortex based on previous protocol[94,104]. In brief, mice were perfused with PBS and kidney cortex was stored in RNAlater for snRNA-seq, whereas for snATAC-seq, kidney cortex without any RNAlater at -80 °C. A 2 mm³ section of tissue was rinsed briefly with PBS and minced before adding 1 mL of lysis buffer containing 20 mM Tris-HCl (pH 8), 320 mM sucrose, 5 mM CaCl₂, 3 mM

MgAc₂, 0.1 mM EDTA, 0.1% TritonX-100, 0.1% RNase Inhibitor and 0.1% DAPI. The tissue was initially dissociated by pipetting up and down 10X with a p-1000 tip and then being passed through a 25 G syringe 10X. The tissue was incubated on ice for 10 min and then passed through a 30 μm CellTrics filter. Nuclei were pelleted by centrifugation (5 min, 500 g, 4 °C) and washed with PBS after removal of supernatant. Nuclei were pelleted again and resuspended in 1 mL of PBS containing 0.04% BSA and RNAse inhibitor (0.2 u/μL) and then passed through a 40 μm FlowMI filter, followed by a 5 μm pluriStrainer Mini filter before generating counts with hemocytometer. Nuclei were then diluted and prepared for snRNA-seq and snATAC-seq with the 10X Chromium System according to the manufacturer's instructions (10X Genomics). Sequencing was performed using a NovaSeqS4 platform.

Raw sequencing data was demultiplexed and aligned to a mouse pre-mRNA reference genome using Cell Ranger 3.1.0-v2 on *SeaWulf*, the HPC Cluster at Stony Brook University. For snRNA-seq, quality control, dimensionality reduction and clustering were performed using the R-package *Seurat 4.3.0*[21]. Genes expressed in a minimum of 3 cells were retained. Cells expressing < 200 or > 7500 genes were excluded. Cells expressing > 5% mitochondrial genes were also excluded. For the snATAC-seq, Cell Ranger atac-2.0.0, *Signac 1.8.0*[20] and *Archr 1.0.2*[22] were used for subsequent analysis. Quality control for each single cell were conducted based on peak_region_fragments>100, peak_region_fragments <60000, pct_reads_in_peaks>5, blacklist_ratio<0.05, nucleosome_signal<4, and TSS.enrichment>1.

For chromatin immunoprecipitation (ChIP)-enrichment analysis, KLF6 binding site data were obtained from ChIP-sequencing data deposited in the Gene Expression Omnibus database (accession no. GSE96355), and locations of binding sites were determined using the Genomic Regions Enrichment of Annotations Tool (GREAT)[105] "basal plus extension" approach, with a maximum extension of 10 kb from any transcription start site (TSS). Heat maps were generated using Morpheus software (https://software.broadinstitute.org/morpheus) and genes were clustered using the one minus Pearson correlation method. Pathway enrichment analyses of the significantly differentially expressed genes were undertaken using *Enrichr*[14,15] libraries of KEGG 2019 (mouse) pathways[18], WikiPathways 2019 (mouse)[17], and Reactome pathways[16]. Trajectory analysis was performed using the R package, Monocle-v2[106,107]. Ligand receptor interactions were shown in the circle plot using R package, *edgebundler 0.1.4*[108]. Pathway enrichment analyses of the genes in the trajectory analysis and for differentially accessible regions (DARs) was carried out using *clusterProfiler 4.6.2*[109,110].

## Proteomics
Tandem mass spectrometry was performed on the supernatant from mouse primary podocytes isolated from *KLF6^PODTA* and *NPHS2-rtTA* mice (with and without doxycycline) using a label-free proteomic approach. After the cells reached confluency, cells were washed five times with 1 × PBS and medium was replaced with phenol-red, insulin-transferrin-selenium (ITS), and serum-free RPMI. After 48 additional hours, the CM harvested and centrifuged for 5 min at 500 g and then for 10 min at 1500 g. Samples were reduced (dithiothreitol), alkylated (iodoacetamide), digested with trypsin, and cleaned using hydrophobic-lipophilic-balance (HLB) pack C-18. For the urine proteomics, the urine samples from *NPHS2-rtTA* and *KLF6^PODTA* mice were spun at 2000 g for 10 min at 4 °C, the supernatant was mixed with a laemmli buffer and heated at 100 °C for 5 min. The protein amount was normalized using creatinine. The samples were run in a 12% criterion TGX gel, the gel was fixed with 40% EtOH/10% acetic acid at room temperature for 15 min. Following a wash with water, the gel was stained with coomassie overnight with gentle rocking. The gel bands were excised based on their molecular weight and cut into small pieces, before undergoing in-gel reduction (dithiothreitol), alkylation (iodoacetamide) and destaining (ammonium bicarbonate/

acetonitrile). The dried gel pieces were digested with trypsin and cleaned through a C-18 column. For both, CM and urine proteomics, samples (4 µl) were injected onto a 20 cm-long ReproSil C-18 (3 µM particle) column and run on the Q Exactive HF Hybrid Quadrupole-Orbitrap Mass Spectrometer (Thermo Fisher Scientific) or TripleTOF 5600+ (Sciex). Analysis was carried out using Thermo Fisher Proteome Discoverer v2.4 and Scaffold.

## Statistical analysis

All statistical analysis was performed using GraphPad Prism 9.0. Based on the distribution of the data, the appropriate parametric or non-parametric statistical tests were utilized. The exact test used for each experiment is denoted in the figure legends and data presented as the mean ± SEM.

## Study approval

All animal studies conducted were approved by the Stony Brook University Animal Institute Committee (#564062). Consent was waived due to the de-identified human biopsy specimens and the study was approved by Stony Brook University Institutional Review Board (#798611). The National Institutes of Health Guide for the Care and Use of Laboratory Animals was followed strictly.

## Reporting summary

Further information on research design is available in the Nature Portfolio Reporting Summary linked to this article.

# Data availability

All data needed to evaluate the conclusions in the paper are present in the paper and/or the Supplemental Materials. Source data are as a separate Source Data file. All raw data from snRNA-seq and ATAC-seq have been deposited in the Gene Expression Omnibus GSE171854 and GSE230681. Raw data from proteomics have been deposited in Massive Database, MSV000092123 [https://massive.ucsd.edu/ProteoSAFe/dataset.jsp?task=08d1896c2de340fca8cf078d9549aa01]. Previously reported datasets used in this study includes accession no. GSE96355. Source data are provided with this paper.

# Code availability

R code and Seurat RDS data object are available through the following link: https://github.com/MallipattuLab/KLF6_PODTA.

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

## Acknowledgements

This work was supported by funds from the National Institute of Diabetes and Digestive and Kidney Diseases Grant (DK112984, DK121846), Veterans Affairs (1I01BX003698, 1I01BX005300) and Dialysis Clinic Inc. to S.K.M. The authors thank Northport Veterans Affairs Medical Center-Stony Brook University Single Cell Genomic Core for single nuclei library generation.

## Author contributions

N.A.G., B.O.F., and S.K.M. designed experiments. N.A.G., B.O.F., and S.K.M. wrote the draft of the manuscript. N.A.G., B.O.F., M.Z., R.B., M.P.R., J.H., I.K., and Y.G. performed experiments and analysis.

## Competing interests

The authors declare no competing interest.
