## [Peer Review File · Nature Communications]

Podocyte-specific KLF6 primes proximal tubule CaMK1D signaling to attenuate Diabetic Kidney DiseaseREVIEWERS' COMMENTS:

Reviewer #1 (Remarks to the Author):

In the manuscript by Gujarati et al, the study examines the role of KLF6 over expressed in podocytes and how this results in clusterin being secreting and acting on the proximal tubule to stimulate the CaMK1D pathway which results in a beneficial effect in STZ, nephrectomy model. The manuscript is well written and these novel findings will have a meaningful significance to nephrology and cell biology in general as they use state of the art tools to dissect this connection.

Some issues that should be addressed experimentally are as follows to strengthen the link between clusterin and CamK1d pathway which appears somewhat limited.

- 1 What are the clusterin expression in glomerulus in DKD
2. An in vivo approach would be important inhibiting either Cam1KD pathway or clusterin with the antibody in the Pod-KLF6 mice to determine whether the beneficial effects are removed.
3. Rescue experiments from supernatant of over expressing clusterin in podocytes would be helpful.
- 4 It is unclear in Supplemental Fig 1b and c why only 20% of KLF6 colocalization with WT1 results in this beneficial effects. A better image should be provided in the main figure for WT1 and KLF6 over expression.
5. Does clusterin interact with Megalin-- by IP or some interaction assay?

Improved images of 7f where the localization is unclear for clusterin

Reviewer #2 (Remarks to the Author):

The authors show that podocyte-specific knockout mouse models of KLF6 have a reduction in albuminuria upon UNx and STZ (uninephrectomy and streptozotocin). This work expands on their initial work demonstrating that KLF6 is protective in this model

<https://pubmed.ncbi.nlm.nih.gov/30115650/>. Now the authors analyze the tubular compartment from the same mouse model using single cell sequencing, and find a potentially injured tubular cell type. They suggest that secreted clusterin might convey information from podocytes to proximal tubules. However, the evidence for the mechanism suggested by the authors is still quite low. Second, some of the experimental decisions taken by the authors remain unclear to this reviewer.

Specific comments:

Figure 3: The majority of PT being “novel” cells requires very careful and extensive benchmarking against previously published datasets describing various stages of injured PTs using single-cell sequencing. This is to make sure that these are not any of the previously described injured or repair cell types/cell states. To claim novelty, careful reanalysis of existing datasets alongside this new data needs to be done.

Figure 5: controls for CAMK1D staining are needed.

Figure 7: The mechanism suggested by the clusterin experiments are not very clean, since effective blocking of the antibody is unknown. Probably recombinant clusterin, or megalin/cubilin inhibition (by knockout or lysine treatment) are needed to demonstrate the cell biological effect on mitochondrial respiration that the authors wish to convey.

Clusterin is an abundant serum protein, and part of it might be derived from serum, not the podocytes. A typical side effect of clusterin is also complement deposition. Can the authors co-stain for complement to get an idea if this is part of general serum protein deposition or a “clusterin only” response? The focus on clusterin is also not very clear given the volcano plot data in Fig. 7c/d.

Reviewer #3 (Remarks to the Author):

This research provides an intriguing exploration of the potential protective role of podocyte KLF6 in diabetic nephropathy. They show that podocyte-specific KLF6 induction triggers the release of secretory clusterin to activate CaMK1D signaling in neighboring PT cells, which attenuates PT dysfunction, and eventual interstitial fibrosis. However, several issues can be identified as follows:

1. In this study, the authors have demonstrated that the overexpression of podocyte KLF6 in a diabetic nephropathy mouse model inhibits the progression of nephropathy. However, it would be beneficial to also show the decrease in KLF6 in diabetic nephropathy podocytes and its mechanisms.
2. Is the increase of podocyte KLF6 a specific therapeutic strategy for diabetic nephropathy, or could it potentially be an effective treatment for other kidney diseases, such as FSGS? In the 5/6 nephrectomy model alone, was the effect of KLF6 not clear? It would be interesting to see observations in other kidney models like ADM nephropathy.
3. It has been reported that there are cases of diabetic kidney disease where renal impairment progresses without showing microalbuminuria. In these cases, podocyte damage may not necessarily precede, and it raises the question of whether supplementing podocyte KLF6 would be an effective treatment strategy.
4. Are there known drugs that can increase podocyte KLF6 or activate the PT CaMK1D signal? For instance, it would be intriguing if ARBs or SGLT2 inhibitors had an effect on these pathways. Do calcineurin inhibitors affect KLF6 expression?
5. Why did the authors choose to perform uninephrectomy (UNx) with subsequent low-dose streptozotocin (STZ) treatment? There are other simple models of diabetic nephropathy without UNx (i.e. single injection of STZ at a high dose (200 mg/kg)).
6. Does clusterin have a protective effect not only on renal tubules but also on podocytes?

7. Due to induction of KLF6 expression in podocytes, chromatin is opened in a site-specific manner according to the results of ATAC-seq. What mechanisms can be considered for this?

8. In the Discussion section, the authors describe that previous studies have shown an increase in clusterin levels in patients with tubular damage. It could be hypothesized that clusterin is secreted in response to podocyte damage to protect against tubular damage. However, why is it insufficient to prevent tubular damage? Could it be that the concentration of clusterin in DKD patients is not sufficiently high to prevent tubular damage?

9. The authors have highlighted the significant roles of KLFs in kidney disease. Is it specific to KLF6 to induce clusterin secretion from podocytes?

Numbered list of new data and analysis experiments:

1. **Fig. 5c:** DEGs in the PT clusters compared to previously reported single cell RNA-seq datasets.
2. **Figs. 7i and j:** OCR in primary PT cells treated with conditioned media from podocytes with and without *ApoJ* overexpression.
3. **Figs. 7n and o:** OCR in primary PT cells treated with recombinant ApoJ +/- cilastatin
4. **Extended Data Fig. 8:** *In vivo* treatment with STO-609 in mice
 - a. PAS, H&E, Picrosirius Red, Lotus lectin, α -SMA immunostaining
 - b. Quantification of Lotus lectin immunofluorescence staining
 - c. Quantification of α -SMA immunofluorescence staining
5. **Extended Data Fig. 5f:** Violin plots for chromatin reorganization associated genes
6. **Extended Data Fig. 7f:** IgG control staining for Camk1d immunofluorescence staining (negative control)
7. **Extended Data Fig. 9:**
 - a. Analysis of *Nephroseq* data for glomerular ApoJ expression from Ju et al.
 - b. Analysis of *Nephroseq* data for glomerular ApoJ expression from Woroniecka et al.
 - c. Immunofluorescence staining for ApoJ, and Synaptopodin
 - d. Immunofluorescence staining for ApoJ, C3-FITC, and C5b-9
8. **Extended Data Fig. 10:**
 - a. *APOJ* expression in *Lenti-ORF* control and *Lenti-ORF APOJ* podocytes
 - b. Western blot for *Lenti-ORF* control and *Lenti-ORF APOJ* podocytes
 - c. *NPHS1* and *Synaptopodin* expression in *Lenti-ORF* control and *Lenti-ORF APOJ* podocytes
9. **Extended Data Fig. 11:**
 - a. Quantification of Lrp2-ApoJ colocalization
 - b-e. Detailed methodology for quantification of Lrp2-ApoJ colocalization
10. Interrogation and quantification of *Klf6* expression in podocytes in previously reported scRNA-seq datasets (**noted in response to reviewer 3 critique**)
11. Interrogation of ApoJ levels in previously reported podocyte proteome and RNA sequencing (**noted in response to reviewer 3 critique**)

Reviewer 1 Comments:

General Comments: “In the manuscript by Gujarati et al, the study examines the role of KLF6 over expressed in podocytes and how this results in ApoJ being secreting and acting on the proximal tubule to stimulate the CaMK1D pathway which results in a beneficial effect in STZ, nephrectomy model. The manuscript is well written and these novel findings will have a meaningful significance to nephrology and cell biology in general as they use state of the art tools to dissect this connection. Some issues that should be addressed experimentally are as follows to strengthen the link between ApoJ and CamK1d pathway which appears somewhat limited.”

Response: We thank the reviewer for the general comments and have addresses the comments below to strengthen the link between ApoJ and CamK1d pathway.

Specific comments:

1. **Comment:** “What are the ApoJ expression in glomerulus in DKD.”

Response: As requested, we interrogated previously reported expression arrays from *Nephroseq* to determine the *ApoJ* expression in glomerular fractions from kidney biopsies with DKD as compared to controls (**Extended Data Figs. 9a-b**). Interrogation of previously reported expression arrays from Ju et al. Genome Res. 2013, PMID: 23950145 and Woroniecka et al. Diabetes 2011, PMID: 21752957 showed an increase in *ApoJ* expression in the glomerular fractions from DKD patients. Additional new co-immunostaining for ApoJ and podocyte marker (synaptopodin) is also shown to demonstrate ApoJ expression in podocytes in **Extended Data Fig. 9c**.

2. **Comment:** “An in vivo approach would be important inhibiting either CaMK1D pathway or ApoJ with the antibody in the Pod-KLF6 mice to determine whether the beneficial effects are removed.”

Response: As requested, we utilized STO-609 (inhibits CaMK1D phosphorylation) in the diabetic *KLF6*^{POD^{TA}} mice to demonstrate that inhibition of CaMK1D signaling exacerbates PT injury and interstitial fibrosis (**Extended Data Fig. 8**).

3. **Comment:** “Rescue experiments from supernatant of over expressing ApoJ in podocytes would be helpful.”

Response: As requested, we overexpressed *ApoJ* in podocytes and treated primary PT cells with conditioned media to demonstrate their protective affect under high glucose conditions (**Figs. 7i-j, Extended Data Fig. 10a-b**).

4. **Comment:** “It is unclear in Extended Data Fig 1b and c why only 20% of KLF6 colocalization with WT1 results in this beneficial effects. A better image should be provided for WT1 and KLF6 overexpression.”

Response: As requested, we improved the immunostaining for WT1 and KLF6 and the updated images are provided in **Extended Data Fig. 1**. Human *KLF6* mRNA expression is uniquely and significantly expressed >50% in **Extended Fig. 1a**, but zinc-finger transcription factors, such as KLF6, have a short protein half-life as they are more likely than other gene families in the human genome to have dosage sensitivity, where small changes have large effects on the transcriptome (Ni et al. Front Genet, 2019, PMID: 31867040).

5. **Comment:** “Does ApoJ interact with Lrp2/megalin-- by IP or some interaction assay?”

Response: Due to the large molecular weight of Lrp2/megalin (~600kDa), co-immunoprecipitation are difficult and, therefore, has not been successfully reported previously in the literature for Lrp2 (Kounnas et al. J Biol Chem 1995, PMID: 7768901). Previous studies have demonstrated that ApoJ undergoes cellular uptake via Lrp2 in other cell types (Zlokovic et al. PNAS 1996, PMID: 8633046; Hamad et al. J Biol Chem, 1997, PMID: 9228033; Bell et al. J Cereb Blood Flow Metab, 2007, PMID: 17077814; Byun et al. EMPO Rep, 2014, PMID: 24825475; Martins Pereira et al. Heart failure reviews 2017, PMID: 28948410; Seo et al. Nat. Commun 2020, PMID: 32371884; de Campos et al. Mol. Biol. Rep 2021, PMID: 34036481, Nielsen et al. KI 2016, PMID: 26759048). Similarly, we demonstrate that ApoJ colocalizes with Lrp2 (**Fig. 7m**), and utilize an unbiased quantitative method the measure the colocalization of Apo-Lrp2 as compared to IgG control (**Extended Data Fig. 11a**). We also show additional images and the detailed methodology used for quantification in **Extended Data Figs. 11b-d**.

6. **Comment:** “Improved images of 7f where the localization is unclear for ApoJ.”

Response: As requested, we redid the immunostaining with colocalization for podocyte marker (synaptopodin) and deconvolution to improve the quality and resolution of the images to ApoJ expression in podocytes (**Fig. 7f, Extended Data Fig. 9c**).

Reviewer 2 Comments:

General Comments: “The authors show that podocyte-specific knockout mouse models of KLF6 have a reduction in albuminuria upon UNx and STZ (uninephrectomy and streptozotocin). This work expands on their initial work demonstrating that KLF6 is protective in this model <https://pubmed.ncbi.nlm.nih.gov/30115650/>. Now the authors analyze the tubular compartment from the same mouse model using single cell sequencing, and find a potentially injured tubular cell type. They suggest that secreted ApoJ might convey information from podocytes to proximal tubules. However, the evidence for the mechanism suggested by the authors is still quite low. Second, some of the experimental decisions taken by the authors remain unclear to this reviewer.”

Response: We thank the reviewer for the comments and have done additional experiments to address the comments and further strengthen the manuscript.

Specific comments:

1. **Comment:** “Figure 3: The majority of PT being “novel” cells requires very careful and extensive benchmarking against previously published datasets describing various stages of injured PTs using single-cell sequencing. This is to make sure that these are not any of the previously described injured or repair cell

types/cell states. To claim novelty, careful reanalysis of existing datasets alongside this new data needs to be done.”

Response: As requested, we interrogated previously reported single cell datasets to confirm that this cluster does not express the injury response genes as described in the previously described “injured” or “repair” PT cluster (**Fig. 5c**). While this suggests that this PT cluster has not been previously reported, we agree with the reviewer that “novel” cells is not the most appropriate label for these group of cells and have renamed it as the “preconditioned-PT” cluster since the cluster is enriched with genes involving canonical pathways in PT cells (i.e., oxidative phosphorylation, TCA cycle, etc.) with the sole induction of KLF6 in podocytes at baseline.

2. **Comment:** “Figure 5: controls for CAMK1D staining are needed.”

Response: As requested, the controls for CAMK1D staining have been added (**Extended Data Fig. 7f**).

3. **Comment:** “Figure 7: The mechanism suggested by the ApoJ experiments are not very clean, since effective blocking of the antibody is unknown. Probably recombinant ApoJ, or Lrp2/megalin inhibition (by knockout or lysine treatment) are needed to demonstrate the cell biological effect on mitochondrial respiration that the authors wish to convey.”

Response: As requested, we conducted both experiments to demonstrate the protective effects of ApoJ using the ApoJ recombinant protein as well as conditioned media from newly generated podocytes overexpressing *ApoJ* (**Fig. 7i-j, 7n-o, Extended Data Fig. 10a-b**). In addition, we also treated PT cells with cilastatin (blocks Lrp2 activity) to demonstrate that protective effects of ApoJ are mediated through Lrp2 (**Fig. 7n-o**).

4. **Comment:** “ApoJ is an abundant serum protein, and part of it might be derived from serum, not the podocytes. A typical side effect of ApoJ is also complement deposition. Can the authors co-stain for complement to get an idea if this is part of general serum protein deposition or a “ApoJ only” response?”

Response: As requested, we stained for C3 and C5b-9 with ApoJ to demonstrate that there is no increase in complement deposition (**Extended Data Fig. 9d**).

5. **Comment:** “The focus on ApoJ is also not very clear given the volcano plot data in Fig. 7c/d.”

Response: As requested, we improved the clarity of ApoJ levels in the volcano plots in **Fig. 7**. In addition, ApoJ is the only uniquely expressed secreted protein in both the podocyte secretome and urine proteome from *KLF6^{POD^{TA}}* as compared to *NPHS2-rtTA* mice (**Fig. 7c-d**).

Reviewer 3 Comments:

General Comments: “This research provides an intriguing exploration of the potential protective role of podocyte KLF6 in diabetic nephropathy. They show that podocyte-specific KLF6 induction triggers the release of secretory ApoJ to activate CaMK1D signaling in neighboring PT cells, which attenuates PT dysfunction, and eventual interstitial fibrosis. However, several issues can be identified as follows:”

Response: We are thankful to the reviewer for reviewing our manuscript and providing us with valuable feedback to strengthen our manuscript. We address each concern point-by-point with additional experiments, data analysis, and reference to previously published data.

Specific comments:

1. **Comment:** “In this study, the authors have demonstrated that the overexpression of podocyte KLF6 in a diabetic nephropathy mouse model inhibits the progression of nephropathy. However, it would be beneficial to also show the decrease in KLF6 in diabetic nephropathy podocytes and its mechanisms.”

Response: We agree with the reviewer and had cited our previously published studies demonstrating that the podocyte-specific knockout of *Klf6* exacerbates DKD by exacerbating mitochondrial injury in the podocytes (Horne et al. Diabetes 2018, PMID: 30115650). We also provide additional details regarding this previous published study in the introduction to provide rationale for the current studies in this manuscript.

2. **Comment:** “Is the increase of podocyte KLF6 a specific therapeutic strategy for diabetic nephropathy, or could it potentially be an effective treatment for other kidney diseases, such as FSGS? In the 5/6 nephrectomy model alone, was the effect of KLF6 not clear? It would be interesting to see observations in other kidney models like ADM nephropathy.”

Response: The role of KLF6 has been previously studied in our lab in other kidney diseases including ADM-induced FSGS nephropathy, demonstrating that induction of KLF6 in podocytes attenuates detrimental effects of adriamycin in cultured podocytes (Mallipattu et al. JCI 2015, PMID: 25689250). We further highlight the details from this study in the main text with supporting references.

3. **Comment:** “It has been reported that there are cases of diabetic kidney disease where renal impairment progresses without showing microalbuminuria. In these cases, podocyte damage may not necessarily precede, and it raises the question of whether supplementing podocyte KLF6 would be an effective treatment strategy.”

Response: We thank the reviewer for raising this important point. Indeed, the results from this study suggest that podocyte-KLF6 preconditions the PT cells against DKD progression and might be protective in the subset of patients with DKD that have primarily PT injury without significant podocyte loss. Furthermore, recent studies suggest that PT injury ultimately determines those individuals that progress in their DKD (Nowok et al. KI 2016, PMID: 26509588; Gilbert et al. Diabetes 2017, PMID: 28325740; Faria et al. Eur J. Pharmacol 2021, PMID: 34303664; Xie et al. KI 2022, PMID: 35469894; Uehara-Watanabe et al. Sci. Rep. 2022, PMID: 35039597). We now added this to the discussion section of the manuscript.

4. **Comment:** “Are there known drugs that can increase podocyte KLF6 or activate the PT CaMK1D signal? For instance, it would be intriguing if ARBs or SGLT2 inhibitors had an effect on these pathways. Do calcineurin inhibitors affect KLF6 expression?”

Response: As requested, we interrogated previously published snRNA-seq dataset from Wu et al. Cell Metabolism 2022, PMID: 35709763, where the diabetic mice were treated with monotherapy or combination therapy of ACEi and SGLT2i for 2 weeks (2 w), db/db were the diabetic mice, db/m was the non-diabetic control, PBS was used as vehicle control. There is no significant changes in podocyte *Klf6* expression in the monotherapy or combination therapy of ACEi and SGLT2i. Furthermore, SGLT2i, such as dapagliflozin and empagliflozin, has been shown to affect CaMK signaling in cardiac myocytes and microvasculature (Mustroph et al. Circulation 2022, PMID: 36251785; Ma et al. Theranostics 2022, PMID: 35836807).

5. **Comment:** “Why did the authors choose to perform uninephrectomy(UNx) with subsequent low-dose streptozotocin (STZ) treatment? There are other simple models of diabetic nephropathy without UNx (i.e. single injection of STZ at a high dose (200 mg/kg)).”

Response: Based on Animal Models of Diabetic Complications Consortium (AMDCC) consortium, single high dose STZ model is not appropriate for DKD changes in mice (Brosius et al. JASN 2009, PMID: 19729434; Breyer et al. JASN 2005, PMID: 15563560). Furthermore, while low- and repeated-dose of STZ induces hyperglycemia, the progression of diabetic nephropathy is very slow and limited with STZ in mice. Therefore, the addition of UNx with STZ accelerates DKD (Uil et al. Scientific Reports 2018, PMID: 29615804) enabling us to determine the protective effects of podocyte-specific KLF6. Conversely, if the goal were to assess an exacerbation in the phenotype, then only STZ can be used. We also utilized this approach

in our previous studies when the low- and repeated-dose STZ-only model was used to show that the podocyte-specific knockout of *Klf6* accelerates DKD (Horne et al. Diabetes 2018, PMID: 30115650). We also tried to generate podocyte-specific KLF6 induction with other genetic diabetic models (such as eNOS^{-/-} + STZ or OVE26 mice), but breeding was challenging due to low fertility with these background strains as previously reported (Drazen et al. Nitric Oxide 1999, PMID: 10534440; Xu et al. AJP-Renal 2010, PMID: 20610531). Rationale has been provided in the manuscript for the choice of model utilized.

6. **Comment:** “Does ApoJ have a protective effect not only on renal tubules but also on podocytes?”

Response: As requested, we reviewed the literature to show that ApoJ has been previously shown to be protective in glomerular diseases (He et al. Sci. Rep. 2020, PMID: 32913257, Saunders et al. KI 1994, PMID: 8196284). This is now added to the discussion section. In addition, induction of *ApoJ* in cultured podocytes showed improved podocytes markers such as nephrin and synaptopodin (**Extended Data Fig. 10c**).

7. **Comment:** “Due to induction of KLF6 expression in podocytes, chromatin is opened in a site-specific manner according to the results of ATAC-seq. What mechanisms can be considered for this?”

Response: As requested, we show that genes involved in chromatin remodeling are differentially enriched in *KLF6*^{POD^{TA}} as compared to *NPHS2-rtTA* mice (**Extended Data Fig. 5f**).

8. **Comment:** “In the Discussion section, the authors describe that previous studies have shown an increase in ApoJ levels in patients with tubular damage. It could be hypothesized that ApoJ is secreted in response to podocyte damage to protect against tubular damage. However, why is it insufficient to prevent tubular damage? Could it be that the concentration of ApoJ in DKD patients is not sufficiently high to prevent tubular damage?”

Response: This is an important point raised by the reviewer and is now addressed in the manuscript. To address this, we interrogated the expression arrays deposited in *Nephroseq* to show that glomerular *ApoJ* expression is increased in later stages as compared to earlier stages of DKD (**Extended Data Figs. 9a-b**). Combined with our mechanistic data, this suggests that direct PT injury under diabetic conditions leads to a loss of Lrp2 receptor activity, which subsequently leads to a lack of utilization of ApoJ, thereby mitigating the increase in CaMK1D-pDrp1 activity (**Fig 7n-o**). Previous studies provide evidence that Lrp2 expression decreases with progression of DKD (Tojo et al. Histochem Cell Biol, 2001, PMID: 11685557; Coffey et al. PLOS one 2015, PMID: 26465605).

9. **Comment:** “The authors have highlighted the significant roles of KLFs in kidney disease. Is it specific to KLF6 to induce ApoJ secretion from podocytes?”

Response: This is an excellent point. As requested, we interrogated our previously reported expression arrays with transgenic mouse models testing podocyte-specific *KLF15* and *KLF4* and did not observe an increase in either glomerular or podocyte ApoJ expression in the podocytes (Guo et al. JASN 2018, PMID: 30143559; Pace et al. Science Advances 2021, PMID: 34516901).

REVIEWERS' COMMENTS

Reviewer #1 (Remarks to the Author):

The authors have adequately addressed the questions the concerns I had in this manuscript to strengthen their findings.

Reviewer #2 (Remarks to the Author):

The authors have responded to all my concerns.

Reviewer #3 (Remarks to the Author):

The revisions have effectively addressed many of the concerns raised in my initial review.

The additional analyses and literature citations have strengthened the manuscript.

While their findings offer a more detailed picture of podocyte KLF6 in diabetic nephropathy, it's worth noting that the importance of KLF6 in DKD podocytes has been previously established through KLF6 knockout studies(Horne et al. Diabetes 2018, PMID: 30115650).

This current work, though valuable, builds upon rather than fundamentally challenges existing knowledge in the field.

Overall, the revisions and additional analyses have resulted in a more robust study that contributes to our understanding of KLF6's role in diabetic nephropathy.